# GENERATIVE ACTOR CRITIC

## ABSTRACT

Conventional Reinforcement Learning (RL) algorithms, typically focused on estimating or maximizing expected returns, face challenges when refining offline pretrained models with online experiences. This paper introduces Generative Actor Critic (GAC), a novel framework that decouples sequential decision-making by reframing *policy evaluation* as learning a generative model of the joint distribution over trajectories and returns, $p(\tau, y)$, and *policy improvement* as performing versatile inference on this learned model. To operationalize GAC, we introduce a specific instantiation based on a latent variable model that features continuous latent plan vectors. We develop novel inference strategies for both *exploitation*, by optimizing latent plans to maximize expected returns, and *exploration*, by sampling latent plans conditioned on dynamically adjusted target returns. Experiments on Gym-MuJoCo and Maze2D benchmarks demonstrate GAC's strong offline performance and significantly enhanced offline-to-online improvement compared to state-of-the-art methods, even in absence of step-wise rewards.

## 1 INTRODUCTION

A central objective in sequential decision-making is to maximize *the expected returns on trajectories* (Sutton et al., 1998), denoted as $\mathbb{E}_{p(\tau)}[Y(\tau)]$, where each trajectory $\tau$ consists of a sequence of states and actions, and $Y$ is a utility function assigning return $y$ to each trajectory. Conventional Reinforcement Learning (RL) algorithms are designed to estimate and optimize this expectation during their training phase. The widely-adopted Actor-Critic framework (Konda & Tsitsiklis, 1999; Haarnoja et al., 2018), for instance, exemplifies this by learning a *critic* to evaluate expected returns and an *actor* to refine the policy towards maximizing these returns. This expectation-centric paradigm is particularly well-suited for online learning settings common in traditional RL (Sutton et al., 1998), due to its amenability to efficient, iterative updates from agent-environment interactions. However, in the context of modern Generative AI (GenAI) pipelines—often characterized by an extensive offline pre-training stage on vast datasets preceding online improvement—we propose to move beyond this expectation-centric approach. We advocate for decoupling the decision modeling process into two distinct phases: (1) train-time generative modeling of the joint distribution over trajectories $\tau$ and their corresponding returns $y$, $p(\tau, y)$, and (2) test-time decision-making, framed as an inference query based on this learned joint distribution. In this paper, we introduce this approach as the Generative Actor-Critic (GAC) framework.

The Generative Actor-Critic framework fundamentally shifts the paradigm from estimating trajectories' expected returns to learning a comprehensive distribution of trajectories and their returns. This holistic approach offers several key advantages. In particular, modeling the distribution $p(\tau, y)$ naturally entails a generative *critic* $p(y|\tau)$ and thus achieves *generalized policy evaluation*; this perspective underscores the ability of GAC to utilize data from various sources when modeling the correlations between behaviors and outcomes, making GAC particularly well-suited for scenarios involving offline pre-training followed by online improvement. Furthermore, by explicitly modeling the entire distribution, GAC can capture complex, multi-modal relationships between behaviors and outcomes—a capacity often limited in methods focusing solely on expectation. While this advantage is shared by distributional RL (Bellemare et al., 2017), the latter typically learns a return distribution conditioned on state-action pairs (often via a distributional Bellman equation) and subsequently falls back to the expectation of this learned return distribution for policy extraction (Bellemare et al., 2017; Barth-Maron et al., 2018; Gruslys et al., 2018). In contrast, GAC enables a fundamental shift that replaces expectation-based *actor* with a versatile inference process on $p(\tau|y)p(y)$. For instance, one

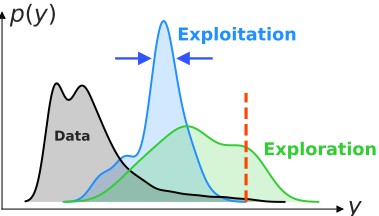

Figure 1: **Conceptual illustration of exploitation and exploration.** After modeling the data distribution (gray), GAC supports distinct test-time inference queries for various decision-making objectives. For *exploitation* (blue), the objective is to maximize the expected return, leading to a focused, low-variance policy that targets high-certainty outcomes. For *exploration* (green), the generative model is conditioned on a target distribution shifted towards higher returns, guiding the search for novel and potentially superior trajectories.

could query the learned model to identify trajectories that maximize expected future returns, similar to policy optimization in distributional RL (Barth-Maron et al., 2018). Alternatively, one could steer the generative process by conditioning on desired high returns to sample corresponding trajectories, akin to mechanisms in Decision Transformer (DT) (Chen et al., 2021; Zheng et al., 2022) and Diffuser (Janner et al., 2022; Ajay et al., 2023). One could also sample diverse yet high-performing trajectories (Lee et al., 2022a) to achieve *generalized policy improvement*.

To operationalize the Generative Actor Critic framework and realize its potential, particularly for complex decision-making tasks and effective offline-to-online improvement, this paper introduces several core methodological contributions: First, we instantiate GAC using a latent-variable model structured as $p(\tau, y, z) = p(\tau|z)p(y|z)p(z)$. Inspired by recent advances like Latent Plan Transformer (LPT) (Kong et al., 2024), a continuous latent plan $z$ is introduced to effectively capture the correlation between high-dimensional trajectories $\tau$ and their corresponding low-dimensional returns $y$. Building on LPT's efficient inference in the latent space, we introduce non-trivial architectural and algorithmic modifications to significantly boost offline performance. Second, leveraging this latent structure, we design designated inference queries for exploitation and exploration. *Exploitation* is formulated as maximizing the expected return $\mathbb{E}[y|z]$ by performing gradient ascent directly in the latent plan space. For *exploration*, we propose to sample $z$ from the posterior $p(z|y^+)$ of target returns from $p(y^+)$, which is slightly shifted from $p(y)$ by a target improvement. Fig. 1 illustrates the intuitions: Starting from the data distribution, exploitation is to refine the policy to focus on a narrow distribution of high-certainty returns, while exploration is to actively seek novel, potentially superior outcomes by shifting the target return distribution towards uncharted, higher-value regions.

Empirically, GAC achieves competitive performance on Gym-MuJoCo and Maze2D benchmarks. Our framework demonstrates strong offline learning capabilities and significantly enhanced improvement in offline-to-online scenarios compared to state-of-the-art baselines, even when step-wise rewards are absent. This strong performance is underpinned by the consistency of our generative components: the actor can be reliably steered by conditioning on target returns, while the critic provides faithful return predictions for given trajectories. Furthermore, our analysis reveals that GAC learns sophisticated internal representations in its latent space, including an implicit world model and a structured cognitive map of the environment, which underpins its strong planning capabilities.

## 2 PRELIMINARIES

**Actor-Critic** The conventional decision-making process in RL is Markov Decision Process (MDP) (Sutton et al., 1998), represented by a tuple $\mathcal{M} = \langle S, A, P, \pi, R, \rho, T \rangle$, comprising a state space $S$, action space $A$, transition function $P : S \times A \mapsto \Pi(S)$, policy $\pi : S \mapsto \Pi(A)$, reward function $R : S \times A \mapsto \mathbb{R}$, initial state distribution $\rho : \Pi(S)$, and horizon $T$. The decision-making objective is to maximize $\mathbb{E}[y]$, *i.e.*, the expectation of *return* $y := \sum_{t=0}^{T} R(s_t, a_t)$ accumulated along the trajectories $\tau = [s_0, a_0, s_1, a_1, \ldots, a_{T-1}, s_T]$. One of the most useful identities in this formulation is the Bellman equation: if we define $Q(s_t, a_t) := R(s_t, a_t) + \mathbb{E}_{P,\pi}[\sum_{k=1}^{T-t} R(s_{t+k}, a_{t+k})]$, we will have $Q(s_t, a_t) = R(s_t, a_t) + \mathbb{E}_{P,\pi}[Q(s_{t+1}, a_{t+1})]$. This Bellman equation, as well as its various variants (Sutton et al., 1998; Ziebart, 2010; Fox et al., 2016; Haarnoja et al., 2018), provides a foundation for learning a value function $Q$ for the expectation of *return-to-go* at each step $RTG_t := \sum_{k=0}^{T-t} R(s_{t+k}, a_{t+k})$. Subsequently, the policy $\pi(a_t|s_t)$ can be updated along an approximated gradient direction $\nabla_{a_t} Q(s_t, a_t)$. That gives us the Actor-Critic framework (Konda & Tsitsiklis, 1999; Silver et al., 2014), where the *critic* $Q$ estimates $\mathbb{E}[RTG_t|s_t, a_t]$, and the *actor* $\pi$ optimizes the expectation from the critic.

**Distributional RL** Bellemare et al. (2017) proposed to shift from the expectation-centric Bellman equation to the distributional version, $RTG_t|s_t, a_t = R(s_t, a_t) + RTG_{t+1}|s_{t+1}, a_{t+1}$. The fundamental difference is that $RTG_t$ is a random variable, while the previous object of interest, $Q_t$, is its mean. Formally speaking, in distributional RL we model $p(RTG_t|s_t, a_t)$ when training the critic, and only calculate the expectation $\mathbb{E}[RTG_t|s_t, a_t]$ when extracting policy for the actor (Bellemare et al., 2017; Barth-Maron et al., 2018; Gruslys et al., 2018). Bellemare et al. (2017) showed that such a generative critic could express the multi-modality of $p(RTG_t|s_t, a_t)$ caused by *state aliasing* (McCallum, 1996), *i.e.*, ambiguity of hidden states under insufficient memory context. However, Bellemare et al. (2017) also formally derived that greedy policy selection (*i.e.* policy optimization toward $\nabla_{a_t}\mathbb{E}[RTG_t|s_t, a_t] = 0$) leads to unstable value iteration and does not guarantee contraction.

**RL via Sequence Modeling** Denoting the policy optimization result as $p(a_t|s_t, Q(s_t, a_t) = Q^*_{s_t})$, where $Q^*$ is the optimal value, we obtain a generative perspective for the actor. This is what underlies the idea of return-conditioned behavior cloning/sequence modeling (Srivastava et al., 2019; Chen et al., 2021; Emmons et al., 2021). Among them, Decision Transformer (Chen et al., 2021) proposed a generalized form of policy $\pi(a_t|s_{\leqslant t}, a_{<t}, RTG_{\leqslant t})$ that is learned with offline data similar to the autoregressive training of language models. However, in test time, since the model does not have access to the $RTG$s anymore, $RTG_0$ is set to be a particular high value (*e.g.*, the maximum or a quantile above the mean) in the offline data, and is updated as $RTG_{t+1} = RTG_t - r_t$ after observing rewards. To set more plausible targets, Lee et al. (2022a) proposed to model not only the action, but also the rewards and $RTG$s. This connects the generative actor $\pi(a_t|s_{\leqslant t}, a_{<t}, RTG_{\leqslant t})$ with the generative critic $p(RTG_t|s_{\leqslant t}, a_{\leqslant t}, RTG_{<t})$ in distributional RL and is closely related to our GAC framework. The difference is that GAC does not assume step-wise rewards, which is argued by Kong et al. (2024) as a more naturalistic setup that results in decision-making models that can do *trajectory stitching* (Fu et al., 2020). We will pinpoint a principled synergy between the generative critic and the generative actor for online policy improvement.

**Offline-to-Online RL** Offline-to-online RL (Wu et al., 2022; Fujimoto & Gu, 2021; Lyu et al., 2022; Kostrikov et al., 2021; Li et al., 2023a; Beeson & Montana, 2022; Zhang et al., 2023; Nakamoto et al., 2023) aims to enhance the sample efficiency of online fine-tuning by leveraging offline pre-training. However, conventional expectation-centric approaches often face challenges during this transition, as their critics can become unreliable due to the action distribution mismatch between the static offline dataset and the evolving online policy (Lee et al., 2022b). Efforts to mitigate this mismatch (Kumar et al., 2020; Nakamoto et al., 2023) can, in turn, inadvertently curtail the exploratory capabilities of the learned actor. Generative actors, such as Decision Transformer, can be fine-tuned using a replay buffer that combines offline and online experiences (Zheng et al., 2022); yet, they often lack stable and scalable mechanisms for targeted exploration in the online phase. In contrast, GAC addresses this gap by providing a distributional perspective for intentional exploration: $p(\tau|y^+)p(y^+)$.

# 3 GENERATIVE ACTOR CRITIC (GAC)

GAC is a framework that separates generative decision-modeling and decision-making as inference. In Section 3.1, we introduce several probabilistic inference queries for different decision-making strategies, motivating the employment of latent-variable generative models. In Section 3.2, we introduce a learning algorithm for GAC based on a latent-variable model (Kong et al., 2024). In Section 3.3 we describe the interplay between the actor, the critic, and the replay buffer (Mnih et al., 2015; Schaul et al., 2015) in realizing exploratory online improvement.

## 3.1 DECISION-MAKING AS INFERENCE

As GAC features a generative model of $p(\tau, y)$, decision-making can be framed as principled inference over it. Previous works (Ziebart, 2010; Botvinick & Toussaint, 2012; Levine, 2018; Abdolmaleki et al., 2018; Qin et al., 2023) in *decision-making as inference* mainly take a stepwise perspective, in which the policy can be viewed as amortized variational distribution $q_{y*}(a_t|s_t)$ for the ground-truth posterior $p(a_t|s_t, y = y^*)$, where $y^*$ denotes the optimal return. However, this posterior, along with its generalized versions in DT (Chen et al., 2021; Zheng et al., 2022), Diffuser (Janner et al., 2022; Ajay et al., 2023), LPT (Kong et al., 2024), is believed to have an *optimism bias* (Ziebart, 2010; Levine, 2018). Consider the decision of whether to buy a lottery ticket, where $y^* = \$10M$ but it is almost impossible to get. The expression $p(a|s, y = y^*)$ asks: what is the distribution over

actions that I should take *given that I have won the lottery*, to which the answer is "buy the winning ticket again", which is overly optimistic. The problem is that the event $y = y^*$ is a counterfactual, so we should not condition on it. Ziebart (2010) attempted to bypass this issue by structuring the variational distribution of $p(\tau|y = y^*)$ as $\prod_0^{T-1} q_{y*}(a_t|s_t)P(s_{t+1}|s_t, a_t)$, where $P(s_{t+1}|s_t, a_t)$ is the ground-truth world dynamics which is not conditioned on the counterfactual.

We want to advocate for a more general solution to optimism bias. The issue is not in the inference process, but in the inference objective. As we all know, the most intuitive decision-making objective is to maximize the expected return $\mathbb{E}[y|\tau]$. And the actual optimal decision "not to buy" is optimal under this objective. Obviously, to a generative model $p(\tau, y)$, $\max_\tau \mathbb{E}[y|\tau]$ is a inference query distinct from $p(\tau|y = y^*)$.

Solving $\max_\tau \mathbb{E}[y|\tau]$ directly would employ gradient ascent that requires costly backpropagation through time. This is where latent modeling excels. Consider a generative model with continuous latent variables $z$

$$p_\theta(\tau, y, z) = p_\alpha(z)p_\beta(\tau|z)p_\gamma(y|z), \tag{1}$$

where conditional independence is assumed between the trajectory $\tau$ and its return $y$, positioning $z$ as an information bottleneck. Intuitively, the latent $z$ are *plans* that abstract trajectories around their returns. As long as $p_\gamma(y|z)$ and $\mathbb{E}[y|z]$ have analytical forms, gradient-based inference can be lifted to the variational posterior $q_\phi(z)$ in the continuous latent space:

$$\max_\phi \mathbb{E}_{q_\phi(z)} \left[ \mathbb{E}[y|z] \right]. \tag{2}$$

Once $z$ is inferred, the policy is entailed by $p_\beta(\tau|z)$. In fact, $z$ can be inferred at any time steps, which is particularly useful when the environment dynamics are stochastic and the policies drift from plans. If we only infer the $z$ at the initial state, we obtain an open-loop policy; if we keep updating $z$ to incorporate the generated partial sequence $\tau_t = [s_0, a_0, s_1, a_1, \ldots, a_{t-1}, s_t]$ via:

$$\max_\phi \mathbb{E}_{q_\phi(z)} \left[ \mathbb{E}[y|z] \right] - D_{\mathrm{KL}}(q_\phi(z)||p_\alpha(z|\tau_t)), \tag{3}$$

where $p_\theta(z|\tau_t) \propto p_\alpha(z)p_\gamma(\tau_t|z)$, we obtain a policy from closed-loop replanning.

Continuous latent modeling also enables other inference queries that would not be imaginable otherwise. In fact, the optimization in Eq. (2) can be viewed as a special type of joint sampling over $p(z, y)$. Returning to the lottery example, a decision policy $p(z|y)$ would not be unreasonable if we had more nuanced control over optimism. In other words, if, instead of sampling from a fixed target $y^*$, we sample from a marginal distribution $p(y^+)$ that is slightly shifted right from $p(y)$, the *optimism* in $p_\theta(z|y^+)$ is tamed with the awareness of $p(y^+)$. To sample from $p_\theta(z|y^+)$, we can employ classical Variational Bayes to $\min_\phi D_{\mathrm{KL}}(q_\phi(z)||p_\theta(z|y^+))$, where $p_\theta(z|y) \propto p(z)p_\gamma(y|z)$, which is equivalent to maximizing the evidence lower bound (ELBO) (Murphy, 2018):

$$\max_\phi \mathbb{E}_{q_\phi(z)}[\log p_\gamma(y^+|z)] - D_{\mathrm{KL}}(q_\phi(z)||p_\alpha(z)). \tag{4}$$

Hopefully, the optimized $q_\phi(z)$ is a faithful approximation of $p_\theta(z|y^+)$, from which we can sample $z$. We will introduce how to sample from $p(y^+)$ in Section 3.3.

## 3.2 GENERATIVE MODELING

LPT (Kong et al., 2024) is an instantion of the latent-variable generative model desribed in Eq. (1). Building on their conceptualization, we made two modifications that align well with some common sense about decision-making: (1) the initial state $s_0$ is not generated from a plan, so we move it into the conditioning; (2) the latent abstractions of the trajectories and their returns are better to be distinct and associated instead of identical, so we reserve one vector $z_y$ for returns, leave the rest $z_{\setminus y}$ for trajectories, and associate them in the prior $p_\alpha(z)$. This gives us the factorization:

$$p_\theta(\tau, y, z) = \rho(s_0)p_\alpha(z|s_0)p_\beta(\tau|s_0, z_{\setminus y})p_\gamma(y|z_y). \tag{5}$$

Following LPT, the prior $p_\alpha(z)$ is implemented as an implicit generative model, but we replace the UNet transformation (Ronneberger et al., 2015) in LPT with a Transformer encoder with bidirectional mask. To sample $z \sim p_\alpha(z)$, we first sample a set of isotropic Gaussian $\epsilon \sim \mathcal{N}(0, I)$ and transform them with the Transformer $z = f_\alpha(s_0, \epsilon)$.

We also inherit LPT's trajectory generator as a $z$-conditioned autoregressive model with a finite context window of size $M$: $p_\beta(\tau|s_0, z_{\backslash y}) = \prod_{t=0}^{T-1} p_\beta(a_t|s_{\leqslant t}, a_{<t}, z_{\backslash y}) p_\beta(s_{t+1}|s_{t-M:t}, a_{t-M:t}, z_{\backslash y})$. This design forces the latent $z$ to serve as global carriers of information, bridging temporal segments that would otherwise be disconnected due to the limited context. The generator is parameterized by a causal Transformer decoder that incorporates the latent $z$ via cross-attention. The stepwise policy and transition distributions are modeled as Gaussians with learnable or fixed variances.

The return predictor models $p_\gamma(y|z_y)$ as a Gaussian $\mathcal{N}(\mu_\gamma(z_y), \sigma_y^2)$, where $\mu_\gamma$ is an MLP that predicts $y$ from the special latent vector $z_y$. The variance $\sigma_y^2$ is either treated as a hyperparameter or learned jointly. Note that $p_\gamma(y|z_y)$ being a Gaussian would not constrain the capability of modeling multi-modal return distribution because $p(y|\tau) = \int p(y|z_y) p(z_y|\tau) \mathrm{d}z_y$ can be very expressive.

For a data point $(\tau, y)$, Maximum Likelihood Estimate (MLE) of latent-variable models is intractable. Instead, we introduce a variational distribution $q_\phi(\epsilon|\tau, y) = \mathcal{N}(\mu, \sigma^2)$, parameterized by $\phi = (\mu, \sigma)$, *i.e.*, the mean vector $\mu$ and a diagonal covariance matrix $\sigma^2$ (Blei et al., 2017; Jordan et al., 1999), and maximize the ELBO:

$$ELBO(\theta, \phi) = \mathbb{E}_{q_\phi(\epsilon)}[\log p_\theta(\tau, y|s_0, \epsilon)] - D_{\mathrm{KL}}(q_\phi(\epsilon)||p(\epsilon)). \tag{6}$$

We use the re-parametrization trick (Kingma & Welling, 2014) in $\mathbb{E}_q$. The training employs alternating update of the local parameters $\phi$ specific to each $(\tau, y)$ with classical Variational Bayes (Hoffman et al., 2013) and the global parameters $\theta = (\alpha, \beta, \gamma)$ shared across all training data. Kong et al. (2025) recently showed that this training scheme is effective at GPT-2 scale language models.

Inference queries in Eq. (2) and Eq. (4) can be adjusted accordingly to incorporate conditioning on $s_0$ and reparametrization with $\epsilon$. Between them, Eq. (2) enables exploitation during test time, while Eq. (4) facilitates exploratory online fine-tuning.

## 3.3 EXPLORATORY ONLINE FINE-TUNING

A key advantage of the GAC framework is that the training objective—maximizing the ELBO in Eq. (6)—remains consistent for both offline pre-training and online fine-tuning. The online fine-tuning is fueled by higher-quality trajectories collected with a principled exploration strategy.

Our exploration strategy is designed to sample trajectories from an improved distribution by targeting returns slightly higher than what the model has previously achieved. Ideally, one might tilt the model's prior $p(z)$ to generate a desired $p(y^+)$. However, we adopt a simpler and more direct approach. We leverage a prioritized replay buffer, initialized with the offline dataset, which serves as an empirical return distribution $p_{\mathrm{data}}(y)$. To generate a target $y^+$ from the target distribution $p(y^+)$, we first sample a high-performing return $y$ from the top-k quantile of the replay buffer. We then add a small, positive increment $\Delta y$ to it, creating an optimistic target $y^+ = y + \Delta y$. This target is used to condition the inference process as described in Eq. (4), yielding latent plans $z \sim p(z|y^+)$ that guide the agent to explore potentially out-of-distribution (OOD) yet superior trajectories.

While the returns collected by the actor $p(z|y)p_{\mathrm{data}}(y)$ from the environment are generally consistent with the target $y$ sampled from the empirical distribution, the introduction of $\Delta y$ creates a mismatch between the target distribution $p(y^+)$ and the ground-truth return distribution. How to reliably set $\Delta y$ to balance exploration with stability is an interesting research question that we leave for future work. In this work, we treat $\Delta y$ as a manually tuned hyperparameter. Empirically, we find that this optimistic distributional targeting strategy effectively guides exploration; while it does not always lead to trajectories that meet the optimistic $y^+$, it consistently outperforms fixed target $y^*$ and gradually shifts the distribution of collected returns toward higher values, as shown in Fig. 2.

The online finetuning process is structured in stages. In each stage, we first collect a set of new trajectories using the exploration policy described above. These newly collected trajectory-return pairs are then added to the replay buffer, enriching the dataset with recent, high-quality experiences. Following data collection, we fine-tune the GAC model on the updated replay buffer for a set number of steps, using the same ELBO maximization objective as in the pre-training phase. This cycle of exploration, replay buffer update, and model fine-tuning is repeated, allowing GAC to progressively adapt and improve its performance through online interaction. The offline pre-training and online fine-tuning algorithms are described by Algorithm 1 and Algorithm 2 in the appendix.

Table 1: **Results on various inference objectives after offline pre-training .** Comparison between the average normalized returns of GAC against several baselines with only final return signals, evaluated without any online finetuning. The mean and standard deviation of different GAC's inference strategies are based on 100 evaluation trajectories. The results illustrate the distinct behaviors among the exploitation query (GAC-$\mathbb{E}[y]$), the exploration query (GAC-$p(y^+)$), fixed target steering (GAC-$y^*$), and sampling from the prior ($p(y|z)p(z)$). The best result in each row is highlighted in bold. In the 1st column, '-m' and '-r' are abbr. for -medium and -replay.

| Dataset | Baselines | | | | | GAC Inference Strategies | | | |
|---|---|---|---|---|---|---|---|---|---|
| | IQL | CQL | DT | QDT | LPT | GAC-$\mathbb{E}[y]$ | GAC-$p(y^+)$ | GAC-$y^*$ | $p(y|z)p(z)$ |
| hopper-m | 35.1 | 23.3 | 57.3 | 50.7 | 58.5 | **67.6**±5.2 | 63.1±8.3 | 60.6±8.0 | 55.8±5.2 |
| hopper-m-r | 13.9 | 7.7 | 50.8 | 38.7 | 71.2 | **83.4**±6.6 | 62.0±22.3 | 39.9±22.8 | 38.8±24.6 |
| walker2d-m | 49.1 | 0.0 | 69.9 | 63.7 | 77.8 | **80.2**±4.6 | 78.9±8.9 | 78.5±6.7 | 75.6±10.1 |
| walker2d-m-r | 5.3 | 3.2 | 51.6 | 29.6 | 72.3 | **78.9**±9.3 | 73.3±10.2 | 76.6±16.2 | 34.3±27.7 |
| halfcheetah-m | 8.5 | 1.0 | 42.4 | 42.4 | 43.1 | **43.6**±2.5 | 41.5±5.1 | 42.5±1.5 | 41.2±5.6 |
| halfcheetah-m-r | 5.2 | 7.8 | 33.3 | 32.8 | 39.6 | **39.8**±7.8 | 38.8±8.9 | 36.7±8.2 | 33.0±12.0 |
| maze2d-umaze | 4.5 | 3.9 | 28.4 | 2.57 | 65.4 | **67.8**±21.4 | 64.2±22.7 | 59.2±21.6 | 35.7±23.4 |
| maze2d-medium | 3.5 | -3.6 | -2.4 | -2.58 | 20.6 | **74.5**±71.0 | 63.3±71.4 | 61.2±70.2 | 19.9±56.6 |
| maze2d-large | 2.0 | -1.2 | -2.5 | -2.51 | 37.2 | **50.3**±40.4 | 39.5±42.3 | 28.9±39.0 | -0.4±6.1 |

## 4 EXPERIMENTS

We evaluate GAC on standard offline and offline-to-online reinforcement learning benchmarks from the D4RL Gym-MuJoCo suite (Halfcheetah, Hopper, and Walker2D) and Maze2D navigation task (Umaze, Medium, Large). The former tasks have dense step-wise reward, while the latter ones feature in binary step-wise reward where the agent is rewarded 1 point when it is around the goal. Our experiments are conducted in a setting where only total trajectory returns are available. The training data contains plenty of suboptimal trajectories. We compare GAC against state-of-the-art methods and analyze the effectiveness of its different inference strategies for exploitation and exploration.

Our baselines come across several paradigms: offline and offline-to-online RL (IQL (Kostrikov et al., 2021), CQL (Kumar et al., 2020), Cal-QL (Nakamoto et al., 2023)), sequence modeling approaches (DT (Chen et al., 2021), ODT (Zheng et al., 2022), QDT (Yamagata et al., 2023), LPT (Kong et al., 2024)), and classic online RL (PPO (Schulman et al., 2017), SAC (Haarnoja et al., 2018)) for reference. Notably, except for LPT, these methods were designed for step-wise rewards. We adapt them to our trajectory-return-only setting for a fair comparison, highlighting the distinct advantage of models that do not rely on dense step-wise reward signals. For online fine-tuning, we use a staged training pipeline, with 100 trajectories for MuJoCo and 500 trajectories for Maze2D each stage.

**Offline Pre-training.** We report the offline pre-training results in the left panel of Table 1, evaluated using the exploitative inference from Eq. (2) (termed GAC-$\mathbb{E}[y]$). Our model consistently outperforms all baselines across the tested environments. GAC's advantage stems from its unique generative approach. Unlike conventional methods rooted in temporal difference learning, GAC forgoes explicit per-step credit assignment. Instead, it learns an implicit understanding of action-outcome relationships via cross-attention across the latent token sequence. This allows the model to learn a smooth and structured latent space of behaviors, enabling it to compose novel, high-quality trajectories by interpolating between successful sub-sequences found in the training data. The robustness of this approach is particularly evident in Maze2D benchmarks, which are characterized by sparse rewards and long-horizon planning challenges. While TD-based methods like IQL and CQL are heavily dependent on dense, step-wise rewards, GAC maintains high performance even with only trajectory-level returns. This highlights a key strength of our framework: GAC achieves superior performance despite operating without the granular, step-wise reward signals that most of these powerful baselines rely on for effective training.

**Inference Strategies.** We analyze the practical implications of the different inference strategies introduced in Section 3.1. We compare three approaches: GAC-$\mathbb{E}[y]$ for pure exploitation via latent space optimization; GAC-$y^*$, which conditions on a standard high-value target return similar to DT (Chen et al., 2021); and GAC-$p(y^+)$, our proposed exploration strategy that conditions on dynamically adjusted targets sampled from the replay buffer.

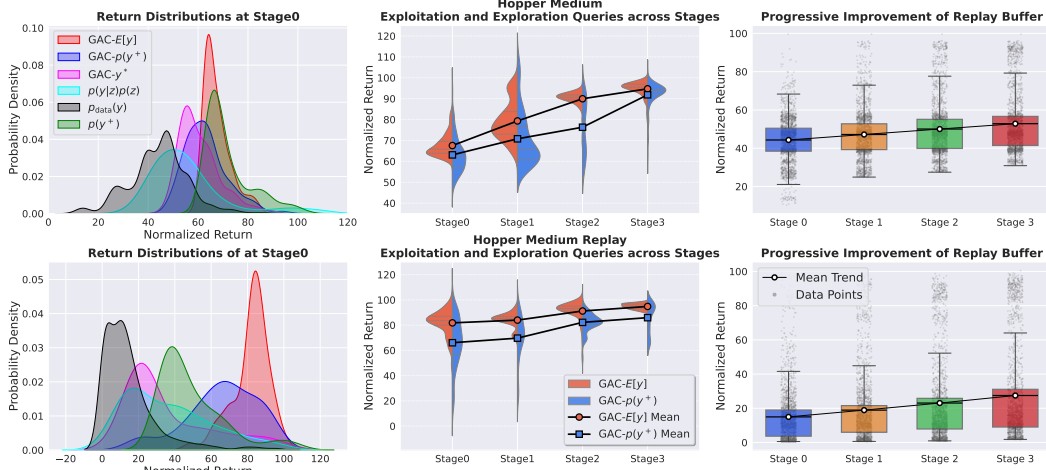

Figure 2: **Illustration of return distributions in Mujoco.** The left column displays the return distributions achieved by various inference objectives after offline pre-training, highlighting the explorative and exploitative behavior. The middle column presents a split violinplot for the inference objectives of GAC-$\mathbb{E}[y]$ and GAC-$p(y^+)$, showing the progressive performance gains and the explorative and exploitative behaviors at three sequential fine-tuning stages. The right column visualizes the progressive improvement in the datasets' quality, as evidenced by the upward shift of the dataset's return distribution over the fine-tuning stages.

The return distributions collected by actors from different inference objectives, visualized in the left column of Fig. 2 (with mean and standard deviation summarized in the right panel of Table 1), highlight their distinct behaviors. The return distribution $p(y|z)$, generated from random prior latent plans from $p(z)$, is dispersed by principle and effectively covers the returns seen in the training data, confirming that the model has not collapsed to a single mode. In sharp contrast, the GAC-$\mathbb{E}[y]$ strategy produces a focused, low-variance distribution concentrated on high-certainty, high-return outcomes, underscoring its efficacy as a pure exploitation method. To encourage exploration, we sample target returns from $p(y^+)$, which apparently have better coverage at high-return and even OOD regions than GAC-$\mathbb{E}[y]$. Based on these targets, GAC-$p(y^+)$ achieves a high mean return with greater variance than GAC-$\mathbb{E}[y]$. While GAC-$p(y^+)$'s optimistic conditioning on $y^+$ leads to a deliberate mismatch between its resulting distribution and $p(y^+)$, it reflects a realistic improvement over the $p_{\text{data}}(y)$, showcasing successful guided exploration toward better outcomes. As GAC-$p(y^+)$'s target distribution is anchored at the data distribution, it also outperforms GAC-$y^*$, whose reliance on a single fixed target (here $y^*$ is the empirical maximum) often leads to overestimation (*optimism bias*) of the outcomes.

**Online Fine-tuning.** Table 2 reports the final returns compared to other methods after fine-tuning. While the gains are relatively small in some tasks, this is often a consequence of its higher initial offline performance. Notably, GAC remains highly effective even without dense, step-wise rewards—a setting where pure online RL methods like PPO and SAC typically underperform. Unlike many offline-to-online methods that rely on explicit optimism or conservatism, GAC improves by leveraging its principled exploration strategy. This strategy systematically expands toward trajectories with higher returns by conditioning on targets set slightly beyond the empirical best of the current dataset. When these exploratory rollouts yield higher-performing trajectories, they are added to the replay buffer. To visualize this dynamic process, Fig. 2 plots the return distributions for MuJoCo from different inference objectives across several fine-tuning stages. The figure clearly illustrates the distinction between exploration and exploitation, tracking the progressive performance improvement via the continuous, positive shift in the collected data's return distribution. As for Maze2D, we illustrate the online improvement through heatmaps in the top row of Fig. 3 which display the mean return achieved from different starting cells across three stages of fine-tuning. We can see a clear progression that from Stage 1 to Stage 3, the number of high-return cells increases, indicating that the agent learns to solve the task from a wider array of initial states. For further details, please refer to Appendix B.

**Emergent Properties.** More significantly, there emerge an abundant set of sophisticated properties, with two identified and illustrated in the bottom row of Fig. 3. First, GAC learns to generate novel, efficient behaviors unseen in the offline data. Despite being trained on suboptimal, axis-aligned trajectories, the fine-tuned model discovers and executes diagonal shortcuts and stops precisely at the goal. This indicates that it has internalized an implicit world model of the environment and reward

Table 2: **Online fine-tuning results.** Comparison of normalized returns before and after online fine-tuning with only access to final return. We fine-tune GAC for 3 stages with data collected by GAC-$p(y^+)$ and report the final GAC-$\mathbb{E}[y]$ performance from 100 trajectories. The best final result for each dataset is highlighted in bold. $\delta$ denotes the performance gain over the offline pre-trained models reported in Table 1.

| Dataset | Online RL | | ODT | | CQL | | Cal-QL | | IQL | | LPT | | GAC | |
|---|---|---|---|---|---|---|---|---|---|---|---|---|---|---|
| | PPO | SAC | online | $\delta$ | online | $\delta$ | online | $\delta$ | online | $\delta$ | online | $\delta$ | online | $\delta$ |
| hopper-m | 13.1±2.1 | 11.2±1.5 | 57.6 | +0.3 | 29.6 | +6.3 | 32.8 | +9.5 | 25.0 | -10.1 | 64.8 | +6.3 | **94.9**±1.9 | +27.3 |
| hopper-m-r | | | 65.2 | +14.4 | 8.4 | +0.7 | 24.1 | +16.4 | 12.6 | -1.3 | 72.4 | +1.2 | **96.8**±13.4 | +14.1 |
| walker2d-m | 9.5±1.6 | 4.0±0.9 | 70.7 | +0.8 | 1.9 | +1.9 | 1.2 | +1.2 | 50.1 | +1.0 | 79.5 | +1.7 | **85.1**±4.9 | +5.4 |
| walker2d-m-r | | | 57.3 | +5.7 | 0.5 | -2.7 | 3.5 | +0.3 | 6.9 | +1.6 | 79.0 | +6.7 | **85.4**±5.3 | +6.5 |
| halfcheetah-m | 1.3±0.1 | 1.7±0.2 | 40.7 | -1.7 | 2.8 | +1.8 | 3.1 | +2.1 | 8.9 | +0.4 | 43.2 | +0.1 | **44.3**±1.1 | +0.8 |
| halfcheetah-m-r | | | 24.4 | -8.4 | 6.4 | -1.4 | 2.3 | -5.5 | 7.4 | +2.2 | 40.6 | +1.0 | **40.9**±0.8 | +1.1 |
| maze2d-umaze | 55.5±1.1 | 61.8±1.2 | 11.2 | -17.2 | 5.8 | +1.9 | 4.9 | +1.0 | 29.3 | +24.8 | 67.2 | +1.8 | **83.5**±23.9 | +15.7 |
| maze2d-medium | 29.3±3.9 | 46.8±4.7 | 2.5 | +4.9 | -1.8 | +1.8 | 0.9 | +4.5 | 23.1 | +19.6 | 26.1 | +5.5 | **166.1**±33.4 | +91.6 |
| maze2d-large | -0.8±0.8 | 17.7±0.8 | 2.2 | +4.7 | 0.3 | +1.5 | 1.1 | +2.3 | 7.4 | +5.4 | 40.1 | +2.9 | **94.2**±42.7 | +43.9 |

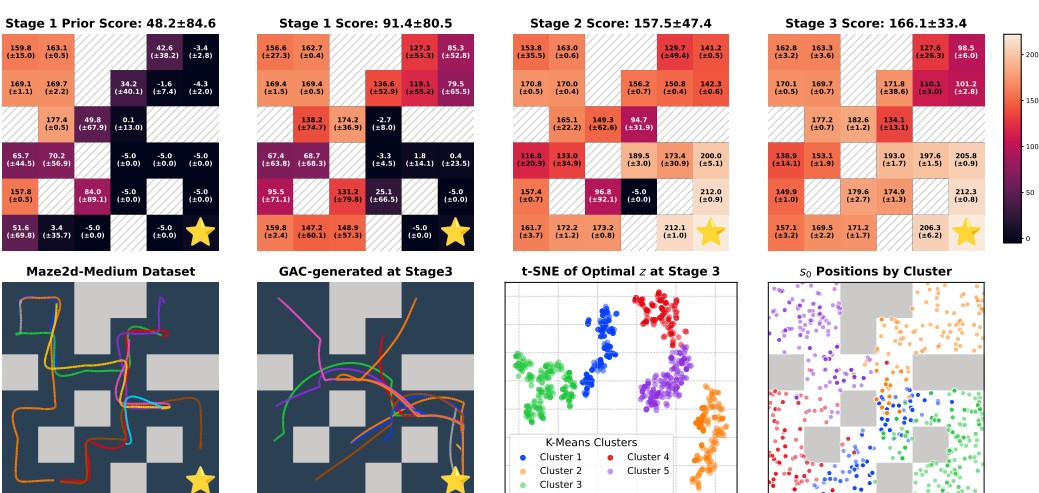

Figure 3: **Illustration of online improvement and emergent properties in Maze.** The top row heatmaps visualize performance from different starting cells. The first two panels compare inference strategies at Stage 1, demonstrating the superiority of the exploitative strategy GAC-$\mathbb{E}[y]$ (Stage 1 Score) over sampling from the prior (Stage 1 Prior Score). The progression from Stage 1 to Stage 3 (second to fourth panels) then illustrates a monotonic improvement in online fine-tuning. Subplots in the bottom row visualize the model's emergent capabilities. The model is trained on suboptimal, axis-aligned trajectories (far left). After fine-tuning, GAC generates novel, efficient diagonal shortcuts (second from left). This advanced planning is likely supported by an emergent cognitive map: a t-SNE visualization shows the latent space of plans, $z$, self-organizing into distinct clusters (third from left), which directly correspond to spatially coherent regions of starting states, $s_0$ (far right).

structure without explicit supervision (Gurnee & Tegmark, 2023; Vafa et al., 2024). Second, GAC's latent space self-organizes into a structured representation of the environment. When we infer latent plans from a dense grid of initial states, they form distinct clusters, with each cluster corresponding to a specific spatial region. This emergent organization is strikingly analogous to hippocampal *place cells* (O'Keefe, 1976; Zhao et al., 2025), which implies a cognitive map (Whittington et al., 2022; Tolman, 1948) is formed, without any spatial priors, to provide a structured foundation for GAC's advanced, long-horizon planning capabilities.

**Actor and Critic.** The success of online improvement is rooted in the efficacy of the actor and the critic learned through generative modeling. To evaluate the consistency of the actor, in Fig. 4 we plot for Hopper-Medium the actual returns achieved by the actor steered by some target returns, which show a strong positive correlation. Depending on the data distributions, actors at different stages behave under different spectrums of variance. Crucially, at Stage 0, the actor can be steered to achieve OOD target returns. Even after Stage 3 when the actor's performance is almost optimal, and exceedingly high target returns won't cause a degradation. There is also a strong correlation between a trajectory's predicted return and the actual return, which justifies the consistency of the critic.

**Closed-loop replanning.** To showcase GAC's replanning capability, we designed a proof-of-concept experiment in Maze2D-Large (Fig. 5). The task requires the agent to navigate a long-range path from a starting state $A$ to a distal goal, passing through an intermediate waypoint $B$. We first observed that an open-loop policy, created by committing to a latent plan $z$ inferred at state $B$, can reliably guide the agent to the goal. However, when starting from $A$, a plan inferred at the start often fails;

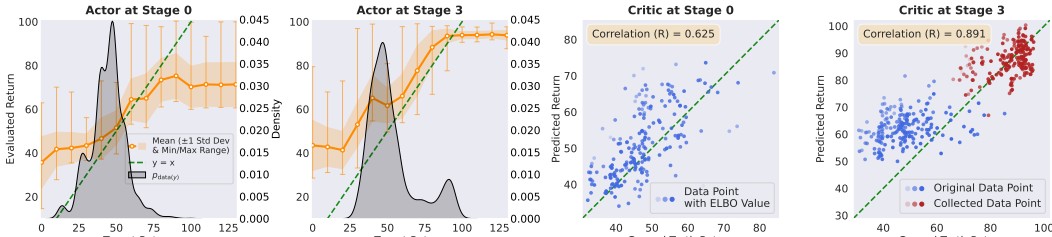

Figure 4: **Analysis of actor and critic.** The left two panels illustrate the actor's consistency. For each target return (x-axis), we infer 50 latent plans to generate corresponding trajectories, and plot the mean of their evaluated returns (y-axis). The right two panels demonstrate the critic's consistency. For each trajectory from the dataset with a ground truth return (x-axis), we infer 50 latent plans and use their average predicted return as the y-axis value. Points with higher ELBO values are less transparent and closer to the ideal $y = x$ line, indicating more reliable predictions. The Stage 3 plot distinguishes original (blue) from newly collected online data (red).

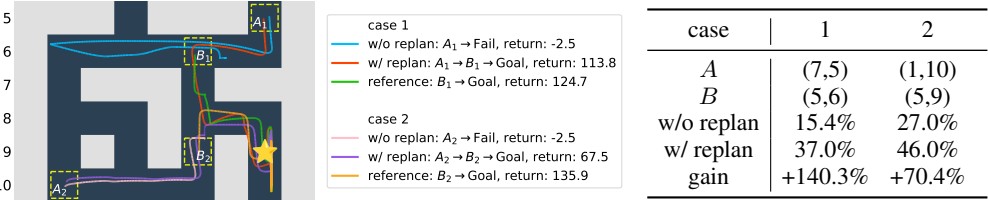

Figure 5: **Experiment on replanning.** While the initial plan from $A$ rarely succeeds, with a replanning at the critical intermediate state $B$, the agent can correct its course towards the goal. In the left we visualize a failure and a correction starting from $A$ and a reference from $B$. In the right we report the success rate.

the agent successfully follows the optimal path to $B$ but then deviates and fails to reach the goal. Our experiment demonstrates that by manually triggering replanning—updating the latent plan $z$ by Eq. (3) upon reaching the vicinity of $B$—the agent can correct its trajectory. This simple act of replanning significantly increases the success rate, highlighting GAC's ability to leverage its generative nature for effective, state-aware self-correction.

**Limitations.** While GAC demonstrates strong general performance, we note that methods with access to step-wise rewards can achieve slightly higher scores in certain tasks (Table 9), such as PPO in Maze2D and Cal-QL in MuJoCo. However, we argue that GAC's performance could become significantly more competitive with the development of a currently unrealized feature: autonomous replanning. Our proof-of-concept experiments show that mid-trajectory replanning substantially improves the success rate, but this capability currently relies on manual triggers. We believe that discovering a principled, autonomous trigger is a promising direction that would unlock GAC's full potential, likely closing the performance gap in complex tasks where open-loop plans are insufficient.

## 5 DISCUSSION

This paper introduced the Generative Actor-Critic (GAC), a framework that shifts the paradigm of reinforcement learning from estimating expected returns to modeling the complete joint distribution of trajectories and their outcomes, $p(\tau, y)$. Our experiments validate this approach, showing that GAC not only achieves state-of-the-art performance but does so by learning remarkably structured internal representations. We argue that the generative objective itself—the need to explain the entire data distribution under the bottleneck of the latents—is what compels the model to move beyond observational pattern matching and form an implicit world model of its environment. The emergence of these properties from a general-purpose learning objective provides a powerful explanation for GAC's strong planning capabilities and suggests that modeling the full distribution might be a key step towards developing agents with a deeper, more fundamental understanding of their world.

The GAC framework orchestrates and extends concepts from several modern RL paradigms. In contrast to Distributional RL, its focus on entire trajectories enables holistic, long-horizon planning. Compared to sequence modeling approaches like Decision Transformer, whose decison-making rely on conditioning with a fixed, potentially *biased* target, GAC offers a more principled approach through its distinct inference strategies for exploitation (GAC-$\mathbb{E}[y]$) and exploration (GAC-$p(y^+)$). While effective, a key limitation of our current approach is the manual tuning of the exploration increment $\Delta y$. Future work should focus on developing adaptive mechanisms to automate this process.

## ETHICS STATEMENT

This work focuses on developing a fundamental algorithmic framework for reinforcement learning. The research is methodological in nature and does not involve human subjects, sensitive personal data, or direct deployment in real-world, safety-critical applications. Our contributions are confined to improving core algorithmic aspects of sequential decision-making, and the experiments are conducted in standard, simulated benchmark environments (Gym-MuJoCo and Maze2D). We do not introduce new datasets that could raise concerns regarding privacy, bias, or misuse. While we recognize that advances in reinforcement learning can have broader societal impacts when applied in downstream applications, our work does not directly engage with these deployment scenarios. The improvements described in this paper are intended for academic research and do not inherently facilitate manipulation, deception, or other unethical uses of AI agents. Overall, we believe that our research poses no direct ethical or societal risks and aligns with the principles of responsible and transparent AI development.

## REPRODUCIBILITY STATEMENT

The findings presented in this paper are supported by a detailed disclosure of our methodology and experimental setup, designed to enable full reproducibility. The core algorithmic formulation of our Generative Actor-Critic (GAC) framework is presented in §3. Our complete experimental protocol, which covers the datasets, evaluation benchmarks, and baselines, is detailed in §4. All implementation details requisite for replication, including model architectures, training procedures, and key hyperparameters, are thoroughly documented in an appendix. Taken together, the paper and its appendix provide a complete blueprint for reproducing our results. We are committed to open science and will release the full source code upon the acceptance of the paper.

## THE USE OF LARGE LANGUAGE MODELS

We used Gemini and ChatGPT as writing assistance tools for language polishing, grammar correction, and improving clarity. These models played no role in research conception, algorithm development, experimental design, or results generation. All scientific content, mathematical derivations, and experimental findings are original author work.

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

## A    RELATED WORK

**Offline RL** algorithms train agents solely from pre-collected datasetsLevine et al. (2020). Most work focuses on reducing extrapolation error from out-of-distribution (OOD) actions. Pessimistic methods lower value estimates for OOD state-action pairs to discourage unreliable actionsYu et al. (2020); Kumar et al. (2020); Kidambi et al. (2020); Luo et al. (2025), while conservative approaches constrain policies to stay close to the behavior policyWu et al. (2019); Peng et al. (2019); Fujimoto & Gu (2021); Fujimoto et al. (2019); Kumar et al. (2019). Compared to develop methods on RL and Bellman equation, our method is more closed to generative modeling to learn a latent plan-conditioned policy (Emmons et al., 2021; Rosete-Beas et al., 2023).

**Exploration** is a major challenge in online RL. Exploration algorithms can be broadly categorized into two groups. For augmented data collection strategies, exploration can be encouraged by action selection perturbations (Painter et al., 2023; Zhou et al., 2020; Wang & Zhu, 2022), guided state selection (Ecoffet et al., 2019), and parameter perturbations (Fortunato et al., 2017). For augmented training strategies, techniques include count-based rewards (Tang et al., 2017; Bellemare et al., 2016; Fu et al., 2017), prediction-based rewards (Burda et al., 2018; Badia et al., 2020b;a), entropy-augmented rewards (Haarnoja et al., 2018), and information-theoretic objectives (Eysenbach et al., 2018; Houthooft et al., 2016). In the LHBG model, exploration and exploitation are balanced by tuning the coefficients of value prediction and KL regularization.

**Offline-to-Online RL** methods aim to bridge the gap between the high exploration cost in online RL and the often suboptimal results of offline RL (Luo et al., 2023). Most existing approaches focus on mitigating the over-conservatism introduced by offline RL algorithms, such as through policy constraints (Kostrikov et al., 2021; Nair et al., 2020; Li et al., 2023a; Beeson & Montana, 2022), policy expansion (Zhang et al., 2023), value calibration (Nakamoto et al., 2023), or value ensembles (Lee et al., 2022b). Some methods introduce generative models to generate data in a more flexible manner to address distribution shift (Liu et al., 2024). Unlike these approaches, our model addresses offline-to-online RL within a generative sequential framework and enables exploratory data collection and online fine-tuning with versatile inference processes after offline pretraining.

**Generative Modeling for Deicision-Making** has emerged as a new paradigm following the introduction of models such as Decision Transformer (Chen et al., 2021) and Trajectory Transformer (Janner et al., 2021). After autoregressive models, diffusion have become influential alternatives for policy learning (Chi et al., 2023) and planning (Janner et al., 2022), with flow matching also gaining attention (Lipman et al., 2022; Zheng et al., 2023). Similar to our method, several latent variable models have been proposed for decision-making. For example, Paster et al. (2022); Yang et al. (2022) and Huang et al. (2023) introduce latent variables to address environmental stochasticity and action multi-modality, respectively. Our approach is closely related to and inspired by Kong et al. (2024), but is unique in its use of an ELBO-based variational Bayes learning and in formalizing various inference objectives especially the one for online data collection. Furthermore, our method differs in specific model architecture and data specification.

**Cross-Entropy Method** (CEM) (Rubinstein, 1999) is a classical example of elite-based sampling methods. It is an iterative optimization algorithm that samples a population of solutions from a parameterized distribution, evaluates them, and then refits the distribution's parameters to the "elite" top-performing subset. This process progressively focuses the search on high-reward regions. While GAC does not implement a standard CEM, its exploration strategy GAC-$p(y^+)$ shares this core principle of iterative, elite-driven refinement. GAC-$p(y^+)$ treats its replay buffer as an empirical distribution of past returns and samples an "elite" set by drawing from its high-performing quantile. Instead of refitting a simple distribution, GAC performs posterior inference to sample latent plans conditioned on optimistic targets derived from this elite set. The resulting high-quality trajectories are added back to the buffer, gradually shifting the collected data distribution toward superior outcomes, effectively achieving a similar goal of iterative population improvement.

**Limitations of step-wise Markov rewards** have been recently discussed by Abel et al. (2021) and further explored by Bowling et al. (2023) and Qin et al. (2023): there are restrictions on what kinds of preferences over policies can be codified in terms of a step-wise reward function that is Markov. In contrast, in the framework of GAC, there is only a final return for a trajectory. This was exemplified by Chen et al. (2021) and adopted by Kong et al. (2024). GAC is the first method that demonstrate performance competitive to methods with step-wise rewards in both offline and online learning.

## B   MORE DETAILS ON EXPERIMENTS

---

**Algorithm 1:** GAC Offline Pre-training

---

1: **Input:** Offline dataset $\mathcal{D}$, model $\theta$, variational params $\phi$.
2: Initialize $\theta$ and $\{\phi_i\}_{i=1}^N$ for all data points.
3: **repeat**
4:     Sample batch $\mathcal{B} = \{(\tau_j, y_j, s_{j,0})\}$ from $\mathcal{D}$.
5:     *// Inner Loop: Local Variational Inference*
6:     **for** each $j \in \mathcal{B}$ **do**
7:         Initialize $\phi_j$. Maximize ELBO (**Eq. equation 6**) w.r.t. $\phi_j$ until convergence to obtain $\phi_j^*$.
8:     **end for**
9:     *// Outer Loop: Global Parameter Learning*
10:     Compute $\nabla_\theta$ELBO using optimized posteriors $q_{\phi*}$.
11:     Update $\theta \leftarrow \theta + \eta_\theta \nabla_\theta$ELBO.
12: **until** convergence of $\theta$.
13: **Return:** Pre-trained parameters $\theta$.

---

**Algorithm 2:** GAC Online Fine-tuning

---

1: **Input:** Pre-trained $\theta$, Env, Buffer $\mathcal{R} \leftarrow \mathcal{D}$, Stages $S$, steps $C, F$, $\Delta y$, top-$k$.
2: **for** stage $s = 1 \dots S$ **do**
3:     *// — Data Collection (Exploration) —*
4:     **for** $c = 1 \dots C$ **do**
5:         $s_0 \leftarrow Env$. Sample base $y$ from top-$k$ of $\mathcal{R}$. Set $y^+ = y + \Delta y$.
6:         Infer plan $z \sim q_\phi^*(z)$ via **Exploration Query** (see Alg. 3) targeting $y^+$.
7:         Generate $\tau \sim p_\beta(\cdot|z, s_0)$, execute, observe $y_{true}$, update $\mathcal{R} \leftarrow \mathcal{R} \cup \{(\tau, y_{true})\}$.
8:     **end for**
9:     *// — Model Fine-tuning —*
10:     **for** $f = 1 \dots F$ **do**
11:         Update $\theta$ on batch from $\mathcal{R}$ maximizing ELBO (as in Alg. 1).
12:     **end for**
13: **end for**

---

**Algorithm 3:** GAC Inference Strategies (Test-time & Exploration)

---

1: **Input:** Query Type, initial state $s_0$, model components $p_\alpha, p_\beta, p_\gamma$.
2: **Switch** Query Type:
3:     **Case Exploitation** ($\max \mathbb{E}[y]$)**:**
4:         *// Active optimization for best expected return*
5:         Find $z^* = \arg\max_z \mathbb{E}_{p_\gamma}[y|z]$ via gradient ascent, initialized from prior $p_\alpha(z|s_0)$.
6:         **Return** $z^*$.
7:     **Case Exploration (Sampling from** $p(z|y^+)$**):**
8:         *// Principled posterior sampling with optimistic target*
9:         Given target $y^+$, solve for $q_\phi^*(z)$ maximizing $\mathbb{E}_q[\log p_\gamma(y^+|z)] - D_{KL}(q\|p_\alpha)$.
10:         **Return** $z \sim q_\phi^*(z)$.
11:     **Case Conditional** ($y = y^*$)**:**
12:         *// Standard "Decision Transformer or Diffuser" style conditioning*
13:         Similar to Exploration, but target $y^*$ is fixed (e.g., max possible return).
14:         **Return** $z \sim q_{\phi, y*}^*(z)$.
15:     **Case Prior Sampling:**
16:         *// Diverse generation without target guidance*
17:         **Return** $z \sim p_\alpha(z|s_0)$.
18: **End Switch**
19: Generate trajectory $\tau \sim p_\beta(\tau|z, s_0)$ using the returned $z$.

---

Inspired by Li et al. (2023b), we employ a more sophisticated training methodology than that of LPT (Kong et al., 2024). This approach aims to tighten the approximation between the Evidence Lower

Table 3: Comparison between Latent Plan Transformer (LPT) and Generative Actor-Critic (GAC).

| Aspect | LPT | GAC (Ours) |
|---|---|---|
| **Latent** $z$ | Monolithic $z$ generates full $\tau$ (including $s_0$). | Decoupled $z_y, z_{\setminus y}$. Prior conditioned on $s_0$. |
| **Initial State** $s_0$ | Generated ($s_0 \sim p(\tau\|z)$); risks mismatch. | Conditioned (Given $s_0$); anchors plan to reality. |
| **Exploitation** | **Conditional Sampling**: $p(z\|y = y^*)$ with fixed target. | **Active Optimization**: Gradient ascent on $\mathbb{E}[y\|z]$. |
| **Exploration** | Stochastic sampling (heuristic); no dedicated mechanism. | **Posterior Sampling**: $z \sim p(z\|y^+)$ with dynamic targets. |
| **Inference** | **MCMC**: Slow convergence, high cost. | **Gradient Ascent**: Fast, stable, lower loss. |
| **Paradigm** | Primarily Offline RL. | Active Online RL (Explore-Collect-Train). |

Table 4: GAC Hyperparameters.

| Architecture-Related | Range |
|---|---|
| Embedding dimension | choice(64, 128, 192, 256, 512) |
| Number of latent tokens | choice(1, 2, 4, 8) |
| Number of attention heads | choice(1, 2, 4, 8) |
| Number of decoder layers | choice(1, 2, 3, 4) |
| Number of encoder layers | choice(1, 2, 3, 4) |
| Context length | choice(1, 4, 16, 32, 64) |

| Architecture-Unrelated | Range |
|---|---|
| Outer-loop learning rate | interval(1e-4, 1e-2) |
| Outer-loop weight decay | interval(1e-4, 1e-2) |
| Outer-loop training steps | choice(1, 5, 10, 20) |
| Inner-loop learning rate | interval(0.01, 0.5) |
| Batch size | choice(800, 700, 600, 500) |

Bound (ELBO) and the log-likelihood, and mitigating the issue of posterior collapse. Specifically, for each data point, we conduct inner-loop optimization until convergence, which is defined by a maximum of 100 training steps and an early-stopping threshold of 1e-4. In the outer loop, unlike the single-step approach in Kong et al. (2024), we treat the number of training steps as a tunable hyperparameter. Given that GAC decouples decision-modeling from decision-making, we can directly assess the quality of our generative model by its loss value, rather than relying on immediate policy evaluation in the environment. Consequently, we leverage the Optuna framework Akiba et al. (2019) for hyperparameter optimization, with the objective of maximizing the ELBO. The complete set of hyperparameters subject to tuning is detailed in Table 4. Following the pre-training phase, architecture-related hyperparameters are fixed, while the remaining parameters are optimized during each subsequent fine-tuning stage. Using the optimal hyperparameters identified (see Table 5 and Table 6[1]), we then configure the inner-loop inference with more stringent convergence criteria: a maximum of 1000 steps and an early-stopping threshold of 1e-5. The offline pre-training and online fine-tuning algorithms are described by Algorithm 1 and Algorithm 2 in the appendix.

For GAC-$p(y^+)$ to collect online data, we introduce another two parameters to obtain a target return set. We perform a data processing procedure on the current return dataset, which we denote as $D$. This process involves a quantile-based filtering and a subsequent transformation. The procedure is

---

[1]The hyperparameters with interval range are fp64 after searching, only displayed four decimal places in Table 6. 'h', 'w', 'ha', 'm', 'r', 'u' and 'l' denotes 'hopper', 'walker2d', 'halfcheetah', 'medium', 'replay', 'umaze' and 'large' respectively.

Table 5: The best GAC Architecture-Related Hyperparameters.

| Parameter | h-m | h-m-r | w-m | w-m-r | ha-m | ha-m-r | m-u | m-m | m-l |
|---|---|---|---|---|---|---|---|---|---|
| Embedding dimension | 192 | 128 | 192 | 192 | 128 | 256 | 192 | 256 | 192 |
| Number of latent tokens | 4 | 8 | 4 | 4 | 8 | 4 | 4 | 8 | 2 |
| Number of attention heads | 4 | 4 | 2 | 8 | 1 | 4 | 1 | 2 | 2 |
| Number of decoder layers | 3 | 4 | 3 | 4 | 3 | 3 | 2 | 2 | 2 |
| Number of encoder layers | 4 | 1 | 3 | 3 | 2 | 2 | 1 | 3 | 2 |
| Context length | 4 | 4 | 1 | 1 | 4 | 1 | 4 | 16 | 4 |

Table 6: The best GAC Architecture-Unrelated Hyperparameters.

| Parameter | h-m | h-m-r | w-m | w-m-r | ha-m | ha-m-r | m-u | m-m | m-l |
|---|---|---|---|---|---|---|---|---|---|
| Stage 0 (pre-training) | | | | | | | | | |
| Outer-loop learning rate | 0.0010 | 0.0003 | 0.0023 | 0.0026 | 0.0026 | 0.0012 | 0.0025 | 0.0002 | 0.0009 |
| Outer-loop weight decay | 0.0002 | 0.0056 | 0.0027 | 0.0011 | 0.0020 | 0.0071 | 0.0005 | 0.0002 | 0.0015 |
| Outer-loop training steps | 10 | 1 | 5 | 1 | 10 | 10 | 5 | 1 | 20 |
| Inner-loop learning rate | 0.2365 | 0.2058 | 0.3980 | 0.0795 | 0.0501 | 0.0452 | 0.0142 | 0.08777 | 0.1172 |
| Batch size | 700 | 500 | 600 | 500 | 500 | 700 | 600 | 500 | 500 |
| Stage 1 (fine-tuning) | | | | | | | | | |
| Outer-loop learning rate | 0.0002 | 0.0018 | 0.0008 | 0.0017 | 0.0062 | 0.0009 | 0.0013 | 0.0034 | 0.0010 |
| Outer-loop weight decay | 0.0002 | 0.0026 | 0.0029 | 0.0062 | 0.0044 | 0.0060 | 0.0008 | 0.0003 | 0.0007 |
| Outer-loop training steps | 1 | 20 | 1 | 20 | 5 | 20 | 1 | 1 | 1 |
| Inner-loop learning rate | 0.0458 | 0.2116 | 0.1223 | 0.0712 | 0.2363 | 0.2836 | 0.0573 | 0.1163 | 0.0712 |
| Batch size | 800 | 700 | 800 | 500 | 700 | 500 | 600 | 500 | 500 |
| Stage 2 (fine-tuning) | | | | | | | | | |
| Outer-loop learning rate | 0.0004 | 0.0010 | 0.0013 | 0.0014 | 0.0093 | 0.0013 | 0.0016 | 0.0028 | 0.0077 |
| Outer-loop weight decay | 0.0006 | 0.0007 | 0.0022 | 0.0050 | 0.0001 | 0.0002 | 0.0006 | 0.0017 | 0.0001 |
| Outer-loop training steps | 1 | 1 | 5 | 20 | 5 | 5 | 1 | 1 | 10 |
| Inner-loop learning rate | 0.1018 | 0.0712 | 0.0124 | 0.1691 | 0.1435 | 0.2154 | 0.2508 | 0.0316 | 0.2089 |
| Batch size | 700 | 500 | 800 | 500 | 600 | 700 | 500 | 800 | 800 |
| Stage 3 (fine-tuning) | | | | | | | | | |
| Outer-loop learning rate | 0.3755 | 0.0005 | 0.0007 | 0.0010 | 0.0039 | 0.0022 | 0.0032 | 0.0017 | 0.0005 |
| Outer-loop weight decay | 0.0077 | 0.0001 | 0.0027 | 0.0007 | 0.0023 | 0.0015 | 0.0030 | 0.0066 | 0.0024 |
| Outer-loop training steps | 5 | 5 | 20 | 1 | 20 | 20 | 1 | 5 | 20 |
| Inner-loop learning rate | 0.0023 | 0.010 | 0.0337 | 0.0712 | 0.1193 | 0.1984 | 0.1030 | 0.2807 | 0.1491 |
| Batch size | 600 | 500 | 600 | 500 | 500 | 600 | 700 | 700 | 500 |

parameterized by two tunable hyperparameters: the quantile threshold, $q$, and an additive constant, $\Delta y$. First, we determine the $q$-th quantile of the dataset $D$, which we denote as $y_q$. All data points in $D$ that are greater than $y_q$ are selected to form a new subset, $D_{\text{sub}} = \{y \in D \mid y > y_q\}$. We finally randomly sample from $D_{\text{sub}}$ and plus $\Delta y$ to obtain the dynamic target return as $\{y + \Delta y \mid y \in D_{\text{sub}}\}$. This procedure allows us to isolate and transform the upper tail of the data distribution, with the specific threshold and transformation intensity controlled by the hyperparameters $q$ and $\Delta y$, respectively. These additional parameters for inference and the standard target return for each environment in Table 7. We present a more complete offline and online results in Table 8 and Table 9, containing outcomes from both step-wise rewards and final return only. We present a We finally present all of the inference results for all Gym-MuJoCo and Maze2D tasks with GAC-$\mathbb{E}[y]$, GAC-$p(y^+)$, GAC-$y^*$ and PRIOR inference strategies in Fig. 7 and Table 10.

Table 7: The Inference Hyperparameters.

| Parameter | h-m | h-m-r | w-m | w-m-r | ha-m | ha-m-r | m-u | m-m | m-l |
|---|---|---|---|---|---|---|---|---|---|
| Standard target return | 3600 | 3600 | 6000 | 6000 | 5000 | 5000 | 165 | 280 | 275 |
| Stage0 $(q, \Delta y)$ | (0.8,10) | (0.8,10) | (0.8,10) | (0.8,10) | (0.6,3) | (0.6,3) | (0.8, 100) | (0.8, 100) | (0.8, 100) |
| Stage1 $(q, \Delta y)$ | (0.8,10) | (0.8,10) | (0.8,10) | (0.8,10) | (0.6,3) | (0.6,3) | (0.8, 100) | (0.8, 100) | (0.8, 100) |
| Stage2 $(q, \Delta y)$ | (0.8,10) | (0.8,10) | (0.6,3) | (0.6,5) | (0.6,3) | (0.6,1) | (0.8, 100) | (0.8, 100) | (0.8, 100) |
| Stage3 $(q, \Delta y)$ | (0.8,10) | (0.8,10) | (0.6,3) | (0.6,5) | (0.6,3) | (0.6,1) | (0.8, 100) | (0.8, 100) | (0.8, 100) |

Table 8: **Offline pre-training results for MuJoCo and Maze2D.** Comparison between the average normalized returns of GAC against several baselines, evaluated without any online finetuning. We assess GAC with the exploitation query, denoted as GAC-$\mathbb{E}[y]$. For each of 5 random seeds, we generate 100 evaluation trajectories and report the mean and standard deviation of their returns. The best result in each row is highlighted in bold.

| Dataset | Step-wise Reward | | | | Final Return | | | | | |
|---|---|---|---|---|---|---|---|---|---|---|
| | IQL | CQL | DT | QDT | IQL | CQL | DT | QDT | LPT | GAC-$\mathbb{E}[y]$ |
| hopper-medium | 63.8 | 58.0 | 60.3 | 66.5 | 35.1 | 23.3 | 57.3 | 50.7 | 58.5 | **67.1**$\pm$0.8 |
| hopper-medium-replay | 92.1 | 48.6 | 63.7 | 52.1 | 13.9 | 7.7 | 50.8 | 38.7 | 71.2 | **81.4**$\pm$1.1 |
| walker2d-medium | 79.9 | 79.2 | 73.3 | 67.1 | 49.1 | 0.0 | 69.9 | 63.7 | 77.8 | **79.3**$\pm$0.9 |
| walker2d-medium-replay | 73.7 | 74.1 | 60.2 | 58.2 | 5.3 | 3.2 | 51.6 | 29.6 | 72.3 | **78.9**$\pm$1.1 |
| halfcheetah-medium | 47.4 | 44.4 | 42.1 | 42.3 | 8.5 | 1.0 | 42.4 | 42.4 | 43.1 | **43.3**$\pm$0.6 |
| halfcheetah-medium-replay | 44.1 | 46.2 | 34.1 | 35.6 | 5.2 | 7.8 | 33.3 | 32.8 | 39.6 | **40.1**$\pm$0.4 |
| maze2d-umaze | 37.7 | 5.7 | 31.0 | 57.3 | 4.5 | 3.9 | 28.4 | 2.57 | 65.4 | **67.1**$\pm$0.5 |
| maze2d-medium | 35.5 | 5.0 | 8.2 | 13.3 | 3.5 | -3.6 | -2.4 | -2.58 | 20.6 | **75.0**$\pm$2.7 |
| maze2d-large | 49.6 | 12.5 | 2.3 | 31.0 | 2.0 | -1.2 | -2.5 | -2.51 | 37.2 | **49.7**$\pm$1.1 |

## C    DETAILED ANALYSIS OF LATENT SPACE STRUCTURE

To address the concern regarding the learned latent structure shown in Fig. 3 being merely (1) an artifact of dimensionality reduction (t-SNE) and (2) a trivial Lipschitz continuous mapping from the initial state $s_0$, we provide a comprehensive quantitative analysis performed in the original latent space, alongside with additional visualization.

**Re-cluster with prior plans.** The clusters visualized in Fig. 3 are optimal $z^*$ from $\arg\max_z \mathbb{E}[y|z]$. To show that the optimization, rather than the initial state $s_0$, is the key factor underlying the clustering, we mix the prior and the optimal plans and redo clustering. The new result is visualized in Fig. 6 where we color the $z^*$ clusters mirroring Fig. 3, and color the additional cluster from $z_{\text{prior}}$ in Brown.

**Optimization shift.** Because the global distance in t-SNE plots are not reliable, we also measure the Euclidean distance between the optimal plan $z^*$ and the initial sample from the prior $z_{\text{prior}} \sim p(z|s_0)$. As shown in Table 11, the average shift is 42.87, which is substantially larger than the average intra-cluster distance 20.53.

**Cluster validity.** Table 11 also compares the average inter-centroid distance and intra-cluster distance, with the former consistently larger than the latter. This confirms that the clusters observed in the t-SNE in Fig. 6 are well separated.

**Pair-wise inter-cluster distances.** We further provide the full pairwise distance matrix between cluster centroids in Table 12. Cluster 3 (Green) and Cluster 2 (Orange) have the maximal distance (53.21) among all cluster pairs, even though their initial states are geometrically adjacent. This separation occurs because the wall between the green and the orange regions warps the task-specific topology away from Euclidean. The model has learned a task-relevant topological representation based on navigational affordance.

Table 9: **Online fine-tuning results.** Comparison of normalized returns before and after online fine-tuning with only access to final return. We fine-tune GAC for 3 stages with data collected by GAC-$p(y^+)$ and report the final GAC-$\mathbb{E}[y]$ performance from five seeds. The best final result for each dataset is highlighted in bold.

| Dataset | Online RL PPO | Online RL SAC | ODT online | ODT δ | CQL online | CQL δ | Cal-QL online | Cal-QL δ | IQL online | IQL δ | LPT online | LPT δ | GAC online | GAC δ |
|---|---|---|---|---|---|---|---|---|---|---|---|---|---|---|
| hopper-m | 13.1±2.1 | 11.2±1.5 | 57.6 | +0.3 | 29.6 | +6.3 | 32.8 | +9.5 | 25.0 | -10.1 | 64.8 | +6.3 | **94.1±1.1** | +27.0 |
| hopper-m-r | | | 65.2 | +14.4 | 8.4 | +0.7 | 24.1 | +16.4 | 12.6 | -1.3 | 72.4 | +1.2 | **95.5±0.8** | +14.1 |
| walker2d-m | 9.5±1.6 | 4.0±0.9 | 70.7 | +0.8 | 1.9 | +1.9 | 1.2 | +1.2 | 50.1 | +1.0 | 79.5 | +1.7 | **84.7±0.3** | +5.4 |
| walker2d-m-r | | | 57.3 | +5.7 | 0.5 | -2.7 | 3.5 | +0.3 | 6.9 | +1.6 | 79.0 | +6.7 | **84.6±0.5** | +5.7 |
| halfcheetah-m | 1.3±0.1 | 1.7±0.2 | 40.7 | -1.7 | 2.8 | +1.8 | 3.1 | +2.1 | 8.9 | +0.4 | 43.2 | +0.1 | **44.3±0.1** | +1.0 |
| halfcheetah-m-r | | | 24.4 | -8.4 | 6.4 | -1.4 | 2.3 | -5.5 | 7.4 | +2.2 | 40.6 | +1.0 | **40.8±0.1** | +0.7 |
| maze2d-umaze | 55.5±1.1 | 61.8±1.2 | 11.2 | -17.2 | 5.8 | +1.9 | 4.9 | +1.0 | 29.3 | +24.8 | 67.2 | +1.8 | **83.5±0.1** | +16.4 |
| maze2d-medium | 29.3±3.9 | 46.8±4.7 | 2.5 | +4.9 | -1.8 | +1.8 | 0.9 | +4.5 | 23.1 | +19.6 | 26.1 | +5.5 | **164.9±0.7** | +89.9 |
| maze2d-large | -0.8±0.8 | 17.7±0.8 | 2.2 | +4.7 | 0.3 | +1.5 | 1.1 | +2.3 | 7.4 | +5.4 | 40.1 | +2.9 | **92.8±0.7** | +43.1 |
| | | | | | | Step-wise Reward | | | | | | | | |
| hopper-m | 77.8±1.3 | 75.0±5.8 | 97.5 | +30.6 | 60.4 | +2.4 | 98.0 | +40.0 | 66.8 | +3.0 | - | - | - | - |
| hopper-m-r | | | 88.9 | +2.3 | 56.3 | +7.7 | 110.0 | +61.4 | 96.2 | +4.1 | - | - | - | - |
| walker2d-m | 43.9±1.4 | 47.5±1.2 | 76.8 | +4.6 | 79.6 | +0.4 | 103.0 | +23.8 | 80.3 | +0.4 | - | - | - | - |
| walker2d-m-r | | | 76.9 | +4.0 | 75.8 | +1.7 | 99.0 | +24.9 | 70.6 | -3.1 | - | - | - | - |
| halfcheetah-m | 25.0±2.6 | 24.4±2.7 | 42.4 | -0.6 | 45.6 | +1.2 | 93.0 | +48.6 | 47.4 | +0.0 | - | - | - | - |
| halfcheetah-m-r | | | 40.2 | +0.4 | 43.0 | -3.2 | 93.0 | +46.8 | 44.1 | +0.0 | - | - | - | - |
| maze2d-umaze | 119.7±1.2 | 115.6±1.3 | 14.0 | -17.0 | 6.0 | +0.3 | 7.4 | +1.7 | 98.6 | +60.9 | - | - | - | - |
| maze2d-medium | 153.4±0.9 | 123.8±1.0 | 4.9 | -3.3 | 8.2 | +3.2 | 8.4 | +3.4 | 62.2 | +26.7 | - | - | - | - |
| maze2d-large | 106.7±9.6 | 96.4±1.4 | 5.1 | +2.8 | 5.1 | -7.4 | 10.6 | -1.9 | 77.6 | +28.0 | - | - | - | - |

Table 10: **Comparing inference objectives for pretrained GAC.** The mean and standard deviation of returns over 100 rollouts from exploitation query (GAC-$\mathbb{E}[y]$), exploration query (GAC-$p(y^+)$), fixed target steering (GAC-$y^*$), and sampling from the prior ($p(y|z)p(z)$).

| Dataset | GAC-$\mathbb{E}[y]$ | GAC-$p(y^+)$ | GAC-$y^*$ | $p(y|z)p(z)$ |
|---|---|---|---|---|
| | | Stage 0 (pre-training) | | |
| maze2d-umaze | 67.8±21.4 | 64.2±22.7 | 59.2±21.6 | 35.7±23.4 |
| maze2d-medium | 74.5±71.0 | 63.3±71.4 | 61.2±70.2 | 19.9±56.6 |
| maze2d-large | 50.3±40.4 | 39.5±42.3 | 28.9±39.0 | -0.4±6.1 |
| | | Stage 1 (fine-tuning) | | |
| maze2d-umaze | 70.7±20.8 | 68.2±21.2 | 62.9±19.8 | 61.6±19.6 |
| maze2d-medium | 91.4±80.5 | 84.9±81.8 | 100.6±78.4 | 48.2±84.6 |
| maze2d-large | 67.4±40.5 | 36.3±43.4 | 32.3±40.5 | 37.6±40.4 |
| | | Stage 2 (fine-tuning) | | |
| maze2d-umaze | 73.1±20.1 | 72.3±22.6 | 70.5±22.1 | 63.0±23.5 |
| maze2d-medium | 157.5±47.4 | 122.7±75.3 | 88.8±80.7 | 84.8±77.4 |
| maze2d-large | 71.7±49.8 | 56.5±56.3 | 55.2±55.5 | 52.2±56.0 |
| | | Stage 3 (fine-tuning) | | |
| maze2d-umaze | 83.5±24.5 | 82.3±24.8 | 72.4±24.8 | 64.0±28.0 |
| maze2d-medium | 166.1±33.4 | 129.1±78.3 | 75.9±89.8 | 41.4±69.0 |
| maze2d-large | 92.5±44.1 | 91.8±44.3 | 84.6±48.4 | 67.6±46.1 |

Table 11: **Euclidean distances in latent space.**

| Metric | Mean Value | Std. Dev |
|---|---|---|
| Optimization Shift ($\|z^* - z_{\text{prior}}\|_2$) | 42.87 | 17.50 |
| Avg. Intra-Cluster Distance | 20.53 | - |
| Avg. Inter-Centroid Distance | 34.25 | - |

Table 12: **Pairwise Euclidean distances between cluster centroids.** Cluster indices correspond to the colors in Fig. 6: 1 (Blue), 2 (Orange), 3 (Green), 4 (Red), 5 (Purple). Note that the **Green-Orange** pair (bolded), despite being geometrically adjacent in the maze, exhibits the largest latent separation due to the wall barrier.

| Cluster | 1 (Blue) | 2 (Orange) | 3 (Green) | 4 (Red) | 5 (Purple) |
|---|---|---|---|---|---|
| 1 (Blue) | 0 | 37.07 | 21.59 | 24.05 | 31.75 |
| 2 (Orange) | 37.07 | 0 | **53.21** | 23.67 | 28.45 |
| 3 (Green) | 21.59 | **53.21** | 0 | 34.38 | 50.52 |
| 4 (Red) | 24.05 | 23.67 | 34.38 | 0 | 37.89 |
| 5 (Purple) | 31.75 | 28.45 | 50.52 | 37.89 | 0 |

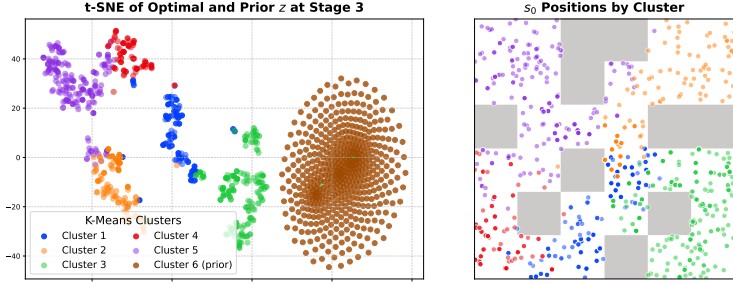

Figure 6: **Visualization of prior and optimized latent plans.** Prior samples form Cluster 6 (Brown), which is clearly separated from optimized latent plans (Clusters 1-5), demonstrating that latent optimization actively drives the latent plans away from the prior distribution to form task-specific clusters.

Figure 7: Complete MuJoCo Results.

