# OpenReview forum: "Generative Actor Critic"
_ICLR.cc/2026/Conference — Submitted to ICLR 2026_

### Official Review · Reviewer_SU1Q · 2025-10-19

**Soundness:** 2
**Presentation:** 2
**Contribution:** 2
**Rating:** 2
**Confidence:** 3

**Summary:**

The submission introduces Generative Actor Critic (GAC), a latent-variable framework that models the joint distribution over trajectories and total returns and treats decision-making as test-time inference on that model. Building on Latent Plan Transformer, the authors split the latent plan into trajectory and return components, train via an ELBO with per-trajectory variational inference, and instantiate separate inference procedures: gradient ascent in latent space to maximize predicted returns for exploitation, and an exploration mode that samples latent plans conditioned on optimistic targets drawn from a replay buffer. Experiments on D4RL MuJoCo and Maze2D benchmarks with only final-return signals report competitive offline performance and staged offline-to-online improvements relative to baselines adapted to the same trajectory-return-only setting; the paper also presents qualitative visualizations (e.g., t-SNE clusters of latent plans) though the interpretive value of those plots is debatable.

**Strengths:**

- The paper tackles the challenging trajectory-return-only setting, demonstrating an approach that does not rely on step-wise rewards while still maintaining competitive performance across several D4RL MuJoCo and Maze2D benchmarks.
- By modeling the joint distribution \(p(\tau, y)\) and performing decision-making as latent-space inference, the framework offers a principled route toward planning and replanning capabilities within a unified generative formulation.
- The separation of exploitation (gradient ascent in latent space) and exploration (conditioning on optimistic targets) provides a clear conceptual structure that could be useful for offline-to-online adaptation scenarios.

**Weaknesses:**

- The paper leans on t-SNE visualizations to argue for a “sophisticated internal representation,” yet $z$ is generated by a network conditioned on $s_{0}$; because standard neural networks are Lipschitz and map nearby inputs to nearby outputs, similar clustering would appear even without meaningful learning, so the claim is unconvincing [1].
- The paper does not quantify the training-time or inference-time overhead introduced by its design choices (e.g., latent optimization, staged fine-tuning), so readers cannot assess whether the method is practical relative to the baselines.
- Exploration depends on adding a hand-tuned $\Delta y$ to top-quantile returns. There is no principled rule for choosing $\Delta y$ or detecting over-optimism, despite large values in Table 6, which is at odds with the Section 3.1 criticism of fixed overly optimistic targets $y^{*}$.
- The paper never specifies how baselines that rely on step-wise rewards (e.g., IQL, CQL, DT) were adapted to the trajectory-return-only setting, so the fairness and reproducibility of the comparison cannot be verified.
- Reported variances (e.g., Table 1) are high, yet there is no analysis of instability or sensitivity; the paper tunes wide ranges of critical hyperparameters (notably $\Delta y$, outer-loop steps, and context length) but provides no sensitivity analysis or ablations, making robustness unclear.
- GAC does not clearly outperform step-wise-reward baselines after fine-tuning, yet the paper provides only qualitative motivation for why modeling $p(\tau, y)$ is preferable, and it omits comparisons to recent strong offline RL methods such as Learning to Trust Bellman Updates [2], leaving the practical advantage in question.

[1] Goodfellow et al., Deep Learning, MIT Press, 2016

[2] Luo et al., Learning to Trust Bellman Updates, arXiv preprint, 2025

**Questions:**

- Could the authors provide ablations on key hyperparameters such as $\Delta y$, the outer-loop training steps, and the context length $M$ to assess sensitivity and robustness?
- How is $\Delta y$ selected in practice, how sensitive are results to this hyperparameter, and how does this choice avoid the optimism issues attributed to fixed $y^{*}$ in Section 3.1?
- Please detail how each baseline that depends on step-wise rewards (e.g., IQL, CQL, DT) was adapted to the trajectory-return-only setting so the comparison can be reproduced.
- Can the authors compare against recent selective regularization methods such as Learning to Trust Bellman Updates [1], or explain why such comparisons are infeasible?
- Reported standard deviations are large for several tasks; can the authors analyze what drives this variance and provide evidence that the results are statistically meaningful?
- Please report wall-clock training time (and online fine-tuning time) for GAC versus each baseline.
- The latent prior $p_{\alpha}(z \mid s_{0})$ already conditions on $s_{0}$, so nearby start states should naturally map to nearby latents (due to Lipschitzness explained above) even without meaningful structure. Could the authors (i) clarify why the t-SNE clusters in Fig.~3(b) are not a trivial consequence of this and (ii) provide additional evidence to substantiate the claim of “sophisticated internal representations”?
- What is the runtime cost of closed-loop replanning (Eq.~3) compared with open-loop execution, especially in the online fine-tuning stages?


[1] Luo et al., Learning to Trust Bellman Updates, arXiv preprint, 2025

---

> ### Author Response · Authors · 2025-11-22
> **Response to Reviewer SU1Q (1/3)**
>
> We sincerely thank the reviewer for their rigorous assessment and valuable suggestions. We especially appreciate your scrutiny of the latent space structure and hyperparameter sensitivity. Addressing your points regarding the t-SNE visualization and the comparison with state-of-the-art Bellman methods has allowed us to provide a much more comprehensive empirical evaluation.
>
> **1. t-SNE resulted from Lipschitz continuity? Unconvincing claim of latent structure?**
>
> We want to clarify a misunderstanding regarding Figure 3, likely caused by our caption. The plot visualizes the optimal latent plan $z^* = \arg\max_z \mathbb{E}[y|z]$ found via gradient ascent, not a simple prior sample $z \sim p(z|s_0)$. Consequently, the reviewer's Lipschitz continuity argument, which applies to direct mappings, does not hold for this optimization solution. Moreover, Figure 3 explicitly refutes the "trivial mapping" claim. Let's focus on the orange and green clusters, which correspond to latent plans starting from spatially adjacent regions separated by a horizontal wall. While these $s_0$ are geometrically proximal in Euclidean space, they are widely separated in z-space, i.e., orange at bottom-right vs. green at far-left in the t-SNE plot. This separation occurs because the wall warps the task-specific topology away from Euclidean.
>
>
> **2. Sensitivity of hyperparameters**
>
> We conduct ablation studies on hopper-medium and maze2d-medium, sweeping context length $M \in \{1, 4, 16, 32, 64\}$ and outer-loop training steps $N \in \{1, 5, 10, 20\}$. Models are trained for 500 steps on Hopper and 1000 steps on Maze2D. We report the final GAC-$p(y^+)$ performance across various $\Delta y$. In the tables below, rows represent the target return gap $\Delta y$, while columns represent the hyperparameter configurations ($M, N$). As shown in table of hopper-medium, the performance is relatively stable column-wise. This indicates that for a fixed configuration (e.g., $M=4, N=10$), GAC remains effective across a reasonable range of $\Delta y$ values, confirming robustness. The model peaks at shorter context lengths ($M=4$). In contrast, table of maze2d-medium shows higher vertical variance. High scores are concentrated in the bottom rows (larger $\Delta y$), validating that larger gaps are needed to filter sub-optimal trajectories. Planning in this task benefits significantly from longer contexts ($M=16$).
>
>
> Hopper-medium performance.
>
> | $\Delta y$ | M1-N1 | M1-N5 | M1-N10 | M1-N20 | **M4-N1** | **M4-N5** | **M4-N10** | **M4-N20** | M16-N1 | M16-N5 | M16-N10 | M16-N20 | M32-N1 | M32-N5 | M32-N10 | M32-N20 | M64-N1 | M64-N5 | M64-N10 | M64-N20 |
> | :--- | :--- | :--- | :--- | :--- | :--- | :--- | :--- | :--- | :--- | :--- | :--- | :--- | :--- | :--- | :--- | :--- | :--- | :--- | :--- | :--- |
> | **1** | 47.1 | 44.9 | 50.2 | 50.1 | 54.8 | 54.9 | 50.7 | 54.7 | 44.9 | 43.3 | 55.7 | 56.4 | 36.8 | 43.4 | 45.4 | 41.7 | 46.3 | 48.3 | 57.2 | 53.3 |
> | **5** | 47.4 | 43.2 | 47.8 | 47.8 | 55.0 | 55.4 | 56.2 | 56.0 | 43.9 | 42.2 | 51.1 | 57.2 | 37.2 | 41.4 | 46.7 | 42.8 | 46.7 | 46.3 | 57.3 | 51.6 |
> | **10** | 48.5 | 48.6 | 45.1 | 50.3 | **58.1** | 60.2 | **63.1** | 57.5 | 44.5 | 43.6 | 53.3 | 57.9 | 37.1 | 43.0 | 51.3 | 42.0 | 48.0 | 49.7 | 53.4 | 53.7 |
> | **20** | 47.8 | 46.6 | 47.3 | 46.2 | 56.9 | 57.3 | **63.5** | 51.2 | 46.3 | 42.3 | 54.3 | 54.8 | 37.4 | 43.9 | 48.9 | 43.1 | 47.7 | 51.3 | 58.0 | 53.9 |
> | **50** | 48.7 | 51.2 | 51.3 | 48.4 | 54.3 | 56.9 | **58.6** | 54.7 | 47.2 | 43.3 | 52.8 | 55.2 | 37.5 | 40.2 | 50.7 | 44.0 | 50.0 | 47.6 | **59.0** | 53.9 |
>
> Maze2d-medium performance.
>
> | $\Delta y$ | M1-N1 | M1-N5 | M1-N10 | M1-N20 | M4-N1 | M4-N5 | M4-N10 | M4-N20 | M16-N1 | M16-N5 | **M16-N10** | M16-N20 | M32-N1 | M32-N5 | M32-N10 | M32-N20 | M64-N1 | M64-N5 | M64-N10 | M64-N20 |
> | :--- | :--- | :--- | :--- | :--- | :--- | :--- | :--- | :--- | :--- | :--- | :--- | :--- | :--- | :--- | :--- | :--- | :--- | :--- | :--- | :--- |
> | **10** | -2.0 | 4.0 | -1.6 | -2.0 | -0.6 | 0.5 | -1.4 | 1.2 | -1.7 | 3.8 | -2.0 | -1.2 | 5.9 | 5.9 | 1.5 | -1.8 | -1.9 | -1.3 | -1.9 | 0.5 |
> | **20** | -2.1 | 4.4 | -2.1 | -2.0 | -2.1 | 16.8 | 17.2 | 3.9 | -4.6 | 1.2 | -1.4 | 4.5 | 5.3 | 0.6 | 8.2 | -0.7 | 16.2 | 8.3 | 9.4 | -0.4 |
> | **40** | -1.9 | 3.9 | -1.6 | -2.1 | 10.1 | 26.2 | 22.1 | 4.0 | 8.2 | 29.1 | 44.1 | 44.3 | 8.6 | 16.8 | 1.4 | 28.2 | 16.0 | 12.5 | -2.9 | 14.2 |
> | **80** | -1.8 | 1.2 | -1.9 | -1.7 | -2.0 | 25.0 | 24.8 | 7.4 | **68.1** | **57.9** | **59.3** | 32.1 | -1.2 | 25.0 | 7.2 | 21.5 | 28.1 | 23.6 | 6.8 | 32.2 |
> | **100** | -1.3 | 7.4 | 0.2 | 8.2 | -0.5 | 6.0 | 24.7 | 4.3 | **63.3** | 55.2 | **62.4** | 42.6 | 10.0 | 26.2 | 26.7 | 22.3 | 21.8 | 19.2 | 12.6 | 19.5 |

---

> ### Author Response · Authors · 2025-11-22
> **Response to Reviewer SU1Q (2/3)**
>
> **3. How to select $\Delta y$ and why it avoids optimism issue.**
>
> We respectfully clarify that our GAC−$p(y^+)$ mechanism is fundamentally different from the fixed $y^\*$ criticized in Section 3.1. Unlike $y^\*$, which is often a static, counterfactual value, our target $p(y^+)$ is a dynamically adjusted distribution. It is derived from the current top-performing parts of the replay buffer, representing "calibrated and achievable optimism" that evolves as the agent improves. We guess that the reviewer’s concern comes from the large $\Delta y$ for maze2d. Actually, this is necessitated by the nature of this specific dataset, where the vast majority of trajectories yield 0 returns. A significant increment is required to push the exploration target beyond this dense, suboptimal mode to incentivize non-trivial pathfinding. Crucially, this aggressive target does not lead to instability because the inference is strictly constrained by the latent prior p(z).
>
> **4. How did we adapt baselines to the trajectory-return-only setting.**
>
> For Value/Policy-based RL Methods (e.g., IQL, CQL, PPO, SAC, Cal-QL): We adapted these methods by treating the problem as a standard sparse-reward MDP. We set all step-wise rewards $R_{t<T} = 0$, the final reward at the terminal step $R_T = y$ (the total trajectory return). For Sequence-Modeling Methods (e.g., DT, ODT): These models are conditioned on a Return-to-Go (RTG) value at each step. In our setting, the true RTG at any timestep $t$ is simply the total, final return $y$. Therefore, we set the RTG condition to $y$ for all timesteps in the trajectory. For LPT, the Latent Plan Transformer framework is already designed to operate on trajectory-level returns, so no adaptation was necessary.
>
> **5. Compare with Luo et al.**
>
> We acknowledge that GAC may slightly lag behind specialized Bellman-based methods when dense, step-wise rewards are available. However, this comparison is not apple-to-apple, pitting GAC’s sparse-return formulation against methods benefiting from dense supervision. Our framework is explicitly designed for the "return-only" setting to bypass the scalability bottleneck of expert-driven reward engineering. Consequently, the slight performance gap in Table 8 is an acceptable trade-off for broader applicability.
>
> To directly address the reviewer's concern about SSAR [1]. We evaluate the Bellman methods in our return-only setting, as we did for IQL/CQL detailed above. The table below shows that state-of-the-art Bellman-based methods fundamentally struggle with the more challenging return-only setting (which prevails LLMs, the up-scaled test field of RL), while GAC excels because it is robust to the absence of step-wise rewards.
>
> | Method | h-m | h-m-r | w-m | w-m-r | ha-m | ha-m-r |
> | :--- | :--- | :--- | :--- | :--- | :--- | :--- |
> | SSAR Cql backbone offline | 12.38 | 19.99 | 3.90 | 14.12 | 36.98 | -0.56 |
> | SSAR Td3+bc backbone offline | 7.35 | 20.27 | 0.27 | 25.48 | -1.64 | -1.28 |
> | GAC-$E[y]$ offline | **67.6** | **83.4** | **80.2** | **78.9** | **43.6** | **39.8** |
> | SSAR Cql backbone online | 14.01 | 20.60 | 7.68 | 13.03 | 37.04 | 0.20 |
> | SSAR Td3+bc backbone online | 14.48 | 20.77 | 0.02 | 5.97 | -2.02 | 1.04 |
> | GAC-$E[y]$ online | **94.9** | **96.8** | **85.1** | **85.4** | **44.3** | **40.9** |
>
> **6. Large standard deviation in maze tasks.**
>
> The reviewer astutely notes the high variance in Table 1. We must clarify that this is not a sign of model instability, but rather an expected statistical artifact of the "all-or-nothing" nature of the sparse-reward Maze2D task. We kindly ask the reviewer to refer to the heatmaps in Figure 3 (top row) and their captions (e.g., "Stage 1 Score: 91.4 ± 80.5"), which perfectly illustrate this. The plots show a clear bimodal pattern of outcomes: from any given starting cell, the agent either succeeds (achieving a high score, ~160) or fails (scoring ~0). The evaluation in Table 1 is an average over all starting cells. When averaging over this bimodal distribution (a mix of 0s and 160s), a high mean score will statistically guarantee a high standard deviation. Therefore, this high variance is not evidence of unclear robustness, but evidence of a difficult, sparse-reward task.

---

> ### Author Response · Authors · 2025-11-22
> **Response to Reviewer SU1Q (3/3)**
>
> **7. Training time**
>
> We acknowledge that the inner-loop optimization in GAC entails a higher computational cost per datapoint compared to standard single-step RL methods. But the total training cost is in fact reasonable. Below provide the current training time with 500 training steps which is already converged, divided into inner-loop(inference) time and outer-loop time. As shown below, training completes in under 6 hours even with a single global update per inferred $z$, with further efficiency gains achievable via more frequent outer-loop optimizations.
>
> | datasets | Total time(s) | Each Inference(s) | Each Outer-loop(s) | Final loss |
> | :--- | :--- | :--- | :--- | :--- |
> | h-m outer-1 | 19548.1 | 31.0 | 0.1 | 0.06393 |
> | h-m outer-5 | 3807.4 | 29.8 | 1.0 | 0.06499 |
> | h-m outer-10 | 2076.7 | 31.9 | 2.1 | 0.06482 |
> | h-m outer-20 | 1038.1 | 31.0 | 4.3 | 0.06631 |
>
> Moreover, this bi-level optimization is an emerging training paradigm in advanced generative modeling, akin to LTM[1]. In fact, [1] has shown signs of comparable training efficiency to standard LLMs. We remain optimistic that the efficiency of this framework can be significantly optimized in future research.
>
> **8. Cost of replanning.**
>
> We didn't thoroughly investigate closed-loop replanning in this work. Generally speaking, the replanning cost can be much lower than initial planning in open-loop execution, because we can warm start from previous plans, which should be closer to the optimal than an inital random guess.
>
> [1] Kong et al., Latent Thought Models with Variational Bayes Inference-Time Computation, ICML, 2025

---

> > ### Comment · Reviewer_SU1Q · 2025-11-24
> >
> > Thank you for the detailed rebuttal and additional experiments; several clarifications were helpful. I am still not convinced on two points:
> > - **t-SNE evidence.** The rebuttal says that Fig.~3 visualizes $z^* = \\arg\\max_z \\mathbb{E}[y|z]$ after gradient ascent, yet the title of the figure still describe ‘t-SNE of prior $z$ inferred from a dense grid of $s_0$. I think they authors should update the caption for clarity, and report how far $z^*$ departs from the prior sample; otherwise the smooth mapping $p(z \\mid s_0)$ continues to explain the clusters. Also note that **t-SNE is non-metric** and routinely produces separated blobs even for random data, so visually distant clusters in the 2-D embedding do not imply the corresponding latents are far apart or meaningful in the original space without quantitative evidence.
> > - **Baseline adaptation.** The rebuttal says step-wise baselines are adapted by setting $R_{t<T}=0$ and $R_T=y$. These algorithms were explicitly designed to learn from batches of individual transitions sampled independently from trajectories, so collapsing the entire return onto the terminal transition eliminates the learning signal for most updates and artificially handicaps the baselines. A fairer adaptation—without introducing any extra reward engineering—would distribute the trajectory return across its transitions (e.g., evenly $y/T$ per step or proportional to timestep) so that each sampled transition carries informative supervision. Please justify why the current setup is appropriate, or revise the adaptation and comparisons accordingly.

---

> > > ### Author Response · Authors · 2025-11-26
> > > **Response to Reviewer SU1Q**
> > >
> > > We thank the reviewer for the prompt response and the continued engagement to improve our work. We value the opportunity to provide further clarifications.
> > >
> > > **Q1. t-SNE and Latent Structure**.
> > >
> > > We appreciate the review’s rigorous scrutiny. We have corrected the typo in the caption. To address the concern on the unreliable global metrics in t-SNE, we also added **Appx C** for a detailed analysis on the original latent space:
> > >
> > > 1. **Reclustering with prior and optimal $z$**: We redo the clustering and visualize the new result in a t-SNE. The prior $z$ indeed forms a new cluster separated from optimal $z$.
> > >
> > > 2. **Significant optimization shift**: We calculated the Euclidean distance between the initial prior ($z_{\text{prior}}$) and the optimal plan ($z^*$). The average shift ($42.87$) is substantially larger than the average intra-cluster distance ($20.53$), indicating that the optimization process significantly refines the plan.
> > >
> > > 3. **Strong cluster validity**: The clusters observed visually are statistically distinct. The distance between cluster centroids is consistently larger than the distance within clusters, confirming that the groupings are well-separated.
> > >
> > > 4. **Confirmed separation of Orange and Green regions**: They are maximally distant in the latent space ($53.21$) among all cluster pairs, confirming that the model has learned navigational affordance rather than just geometric proximity.
> > >
> > > **Q2. Baseline adaptation.**
> > >
> > > 1. **Empirical evaluation of evenly distributed rewards**: We follow the reviewer’s suggestion ($r_t = y/T$) to adapt SSAR and DT, which result in mixed results. The adaptation succeeded in MuJoCo but caused significant performance degradation in Maze2D. This further justified that SOTA RL methods may be optimized towards a specific class of stepwise rewards, to which GAC is agnostic.
> > >
> > > 2. **Justification for the current setup**: Our setup $R_{t<T}=0,R_T=y$ follows the standard "Delayed Reward" benchmark established by Decision Transformer. This setting is widely adopted to rigorously test long-horizon credit assignment capabilities without the crutch of dense, step-wise supervision. Furthermore, learning from final outcomes rather than step-wise process rewards aligns with emerging trends in RL-based reasoning and Large Language Models (e.g., DeepSeek-R1 [1]). In these domains, valid step-wise rewards are often unavailable, making the ability to learn solely from trajectory-level outcomes a critical capability.
> > >
> > > | Method | maze-umaze | maze-medium | maze-large | h-m | h-m-r | w-m | w-m-r | ha-m | ha-m-r |
> > > | :--- | :--- | :--- | :--- | :--- | :--- | :--- | :--- | :--- | :--- |
> > > | SSAR Cql backbone offline | -17.3 | -5.0 | -2.5 | 47.5 | 94.7 | 78.3 | 81.2 | 65.7 | 37.4 |
> > > | SSAR Td3+bc backbone offline | -17.3 | -5.0 | 38.4 | 89.2 | 64.6 | 88.7 | 51.6 | 59.0 | 48.7 |
> > > | Decision Transformer | -17.3 | -5.0 | -2.5 | 64.7 | 71.6 | 71.3 | 70.0 | 40.7 | 38.9 |
> > > | GAC-$E[y]$ offline | 67.8 | 74.5 | 50.3 | 67.6 | 83.4 | 80.2 | 78.9 | 43.6 | 39.8 |
> > > | SSAR Cql backbone online | -17.3 | -5.0 | 13.1 | 102.4 | 96.6 | 84.8 | 85.1 | 78.4 | 51.0 |
> > > | SSAR Td3+bc backbone online | -17.3 | -5.0 | 44.1 | 100.7 | 94.8 | 91.9 | 96.4 | 80.7 | 65.3 |
> > > | Online Decision Transformer | -17.3 | -5.0 | 9.1 | 93.9 | 77.4 | 72.4 | 71.5 | 42.0 | 40.0 |
> > > | GAC-$E[y]$ online | 83.5 | 166.1 | 94.2 | 94.9 | 96.8 | 85.1 | 85.4 | 44.3 | 40.9 |
> > >
> > > [1] DeepSeek-AI, Daya Guo, et al., DeepSeek-R1: Incentivizing reasoning capability in LLMs via reinforcement learning, arxiv, 2025

---

### Official Review · Reviewer_xAm5 · 2025-10-27

**Soundness:** 4
**Presentation:** 3
**Contribution:** 4
**Rating:** 8
**Confidence:** 3

**Summary:**

This manuscript provides an interesting framework Generative Actor-Critic to decouple the decision-making process. In the policy evalution step, the authors learn a generative distribution for the joint distribution of trajectory and returns, instead of the conventional mean return. For the model inference, the authors suggests a novel exploration and exploitation method. The offline experiments from MuJoCo and Maze2D indicate superiority of the proposed method.

**Strengths:**

* The first strength of this manuscript is the joint distribution estimation for trajectory and return, which differs from the  method in traditional RL, and this may be particularly suitable for long horizon planning.

* The online fine-tuning step in GAC is critical. The exploration idea of $y^{+} = y+\Delta y$ seems an efficient way for better exploration.

* The idea of applying the latent variable in generative modeling is appealing. It combines the good merits in generative modeling (dimension reduction and re-parametrization), and applies well in exploration and exploitation.

**Weaknesses:**

1. Some key hyper-parameters relies on manual tuning, and the robustness of the proposed method needs to be improved. For example, the $\Delta y$ is important for online fine tuning, however, the authors have not proposed the auto-tuning method. This also applies for the hyper-parameters in latent variable modeling.

2. The reviewer appreciate the idea of modeling the joint distribution of trajectory and returns. However, it would be better if additional theoretical analysis are provided. Why this idea could work and how does it perform in convergence, especially why it could avoid the traditional Out-Of-Distribution issue in RL.

3. The idea in the manuscript is good. However, the model design, which includes the ELBO optimiation, generative modeling (latent variable) and online fine-tuing, the cost of model training should be much high the conventional RL method. Have the authors consider about this issue?

**Questions:**

See the weakness above

---

> ### Author Response · Authors · 2025-11-22
> **Response to Reviewer xAm5**
>
> We are grateful for the reviewer’s constructive feedback, particularly regarding the robustness of hyperparameters and training efficiency. Your insights prompted us to conduct additional ablation and provide a detailed breakdown of computational costs, which have strengthened the practical evaluation of our work.
>
> **1. Hyperparameter tuning**
>
> To address the concern regarding robustness, we have included results for different $\Delta y$ with a wide range of values on the Hopper and Maze2D dataset. We can see that while optimal $\Delta y$ varies by tasks, there is always a range of $\Delta y$ for each task that results in comparable perfomance. But we do agree that automating $\Delta y$ (similar to entropy tuning in SAC) is a promising direction for future work.
>
> | $\Delta y$ | 1 | 5 | 10 | 20 | 50 |
> | :--- | :--- | :--- | :--- | :--- | :--- |
> | **h-m** | 50.7 $\pm$ 8.1 | 56.2 $\pm$ 9.2 | 63.1 $\pm$ 8.3 | 63.5 $\pm$ 10.1 | 58.6 $\pm$ 11.3 |
> | **h-m-r** | 37.2 $\pm$ 22.7 | 45.1 $\pm$ 21.5 | 62.0 $\pm$ 22.3 | 59.1 $\pm$ 19.4 | 52.5 $\pm$ 24.8 |
> | **$\Delta y$** | **10** | **20** | **40** | **80** | **100** |
> | **m-m** | -1.7 $\pm$ 7.3 | -4.6 $\pm$ 1.8 | 8.2 $\pm$ 33.2 | 68.1 $\pm$ 69.2 | 63.3 $\pm$ 71.4 |
> | **m-l** | -2.0 $\pm$ 1.9 | 0.6 $\pm$ 7.0 | 35.3 $\pm$ 30.7 | 27.4 $\pm$ 42.1 | 39.5 $\pm$ 42.3 |
>
> The reviewer can also check the requested results on different context length and outer-loop training steps in "Response to Reviewer SU1Q".
>
> **2. Why the online improvement could work and why it avoids OOD issues?**
>
> As a generative model, GAC's online improvement would continue as long as the training data is getting better. And the data is getting better because the replay buffer is updated to only incorperate better samples from exploration. Then the question is reduced to why GAC's exploration can avoid the OOD issue and obtain better samples. The intuition is that GAC follows a probabilistically modular design rooted in Bayesian principles, where latent plans follow an explicit learned prior $p(z)$. In exploration, this prior $p(z)$ acts as an explicit regularizer in the posterior update. It counterbalances the optimistic likelihood $p(y^+|z)$, filtering out plans that theoretically satisfy the high target return but are structurally implausible. In contrast, conventional RL and conditional generative models (e.g., Decision Transformer/Diffuser) lack these explicit representations of prior or return uncertainty to constrain their decision boundaries. While we plan to formalize this intuition in future work, our current empirical results—specifically the Actor's stability under OOD targets shown in Figure 4—provide strong novel evidence for the effectiveness of this Bayesian mechanism.
>
> **3. Training cost**
>
> We acknowledge that the inner-loop optimization in GAC entails a higher computational cost per datapoint compared to standard single-step RL methods. But the total training cost is in fact reasonable. Below provide the current training time with 500 training steps which is already converged, divided into inner-loop(inference) time and outer-loop time. As shown below, training completes in under 6 hours even with a single global update per inferred $z$, with further efficiency gains achievable via more frequent outer-loop optimizations.
>
> | datasets | Total time(s) | Each Inference(s) | Each Outer-loop(s) | Final loss |
> | :--- | :--- | :--- | :--- | :--- |
> | h-m outer-1 | 19548.1 | 31.0 | 0.1 | 0.06393 |
> | h-m outer-5 | 3807.4 | 29.8 | 1.0 | 0.06499 |
> | h-m outer-10 | 2076.7 | 31.9 | 2.1 | 0.06482 |
> | h-m outer-20 | 1038.1 | 31.0 | 4.3 | 0.06631 |
>
> Moreover, this bi-level optimization is an emerging training paradigm in advanced generative modeling, akin to LTM[1]. In fact, [1] has shown signs of comparable training efficiency to standard LLMs. We remain optimistic that the efficiency of this framework can be significantly optimized in future research.
>
> [1] Kong et al., Latent Thought Models with Variational Bayes Inference-Time Computation, ICML, 2025

---

### Official Review · Reviewer_UgEN · 2025-10-31

**Soundness:** 4
**Presentation:** 4
**Contribution:** 3
**Rating:** 8
**Confidence:** 3

**Summary:**

The authors introduce the Generative Actor-Critic (GAC), a framework that learns the joint distribution between trajectories and returns, in contrast to most prior work that focuses on generating trajectories that simply maximize returns. GAC differs from LPT in both its parameterization and training procedure:
- GAC generates a plan for each initial state.
- GAC uses separate conditioning latents for trajectories and returns.
- GAC learns an approximate posterior using variational Bayes, rather than MCMC as in LPT.

The authors further leverage their framework to design a novel inference and data collection strategy, in which the trained model is conditioned on a translated version of the return to guide it toward high-reward regions. This approach enables the model to produce high-quality samples that differ from those in the offline dataset, as demonstrated in Figure 4.

Additionally, GAC supports closed-loop re-planning, with promising results shown in Figure 5. Overall, the proposed method outperforms baselines both after offline pre-training and subsequent online fine-tuning. The actor and critic derived from the learned joint model exhibit good alignment with the ground-truth returns, and the approach demonstrates versatility when conditioned on the initial state, as illustrated in Figure 3.

**Strengths:**

The paper is well-written and analyzes the proposed approach holistically on many aspects and does so very well with convincing evidence. The paper also well motivates the problem and it's easy to follow the motivation all the way through the proposed solutions. I also liked the analysis done in Figure 3 that intuitively shows the effect of online-finetuning as well as Figure 4 which analyzes the quality of the actor and the critic. Figure 5 also motivates a natural extension and future work by showing the potential for integrating closed-loop re-planning during inference.

**Weaknesses:**

- As pointed out by the authors, GAC and LPT both suffer from the inability to incorporate step-wise rewards which should be used whenever available and Table 8 reflects the consequence of not incorporating them. That being said, it would be interesting to see a heuristic for autonomous re-planning that can for example incorporate those step-wise rewards to "mine" for interesting states and use them.
 - On one hand, the GAC paper builds on top of the LPT paper and proposes some modifications to its core architecture and training. On the other hand it still introduces some novel ways for conducting inference. Looking at Table 1, it seems that the architectural and training changes actually perform worse than LPT when we look at the p(y|z)p(z) column which calls into question the proposed GAC pre-training scheme and its architecture. This also begs the following question: what happens when we keep LPT yet use the inference schemes from GAC? I propose the following ablations:
	 - Inference strategies: Keep the core LPT training and architecture but compare against the same introduced inference strategies, namely: p(y+), E[y] and y*. Also, in LPT they use Exploitation-inclined inference (EI) which I see it as close to p(y+) and E[y] so it would also be interesting to highlight results for that inference scheme for both LPT and GAC.
	 - Training:
		 - What happens to GAC when you don't split the conditioning vector?
		 - What happens when the plan doesn't depend on $s_0$?
		 - What happens when you use MCMC instead of a variational approximation to the posterior?

**Questions:**

- In line 189, do you mean $p_\alpha$?
- Since the paper modifies the LPT, it would be good to have a table summary summarizing differences
- It would be good to know how fast is this approach compared to existing methods. LPT uses MCMC to get the posterior samples that
- Would be good to have the algorithms for the other inference strategies
- it would be good to highlight the second and third best just to be able to easily see how other inference strategies fare with the baselines
- In Table 2, it would be good to indicate what delta is in the caption. I understand it to be the difference between the returns after the online-finetuning and the returns from the offline-pretrained model. Also, some cells in the table give the sign for delta and some do not. I suggest you standardize that over all cells where it applies.
- line 424 typo: alomst -> almost
- How is re-planning done exactly? Is the latent plan generation triggered from the partial trajectory from A to B whenever the agent gets to the waypoint B?

---

> ### Author Response · Authors · 2025-11-22
> **Response to Reviewer UgEN**
>
> We are deeply grateful to the reviewer for the thorough, insightful, and highly constructive feedback. We are especially thrilled by your recognition of GAC as a novel RL framework and its potential to reframe decision-making. Your insightful questions, particularly regarding the use of step-wise rewards, are exceptionally valuable. As we detailed in our response, we see this as an excellent direction for future work and are excited to prioritize it.
>
> **1. Incorporating step-wise rewards**
>
> We completely agree that bridging the gap with methods using step-wise rewards is a vital direction. As demonstrated in our Figure 5 proof-of-concept, GAC's generative nature already supports powerful state-aware replanning, which drastically boosted success rates (e.g., +140.3% in Maze2D). In this context, we find your suggestion to utilize step-wise rewards particularly valuable as a potential mechanism for autonomous replanning. For instance, we can trigger a re-plan when immediate rewards are abnormally low.
>
> **2. GAC v.s. LPT**
>
> - **Prior sampling of GAC underperforms LPT?**
>
> We want to clarify that the observed performance difference stems from a mismatch in inference objectives. The reported LPT results utilize conditional sampling with high target returns (equivalent to our GAC$-y^*$ baseline), whereas the GAC figures reflect unconditioned prior sampling ($p(y|z)p(z)$). These queries serve distinct purposes: the prior is designed to capture the full diversity of the dataset rather than solely maximizing returns. As shown in Figure 2 (left), our prior sampling successfully covers a wide spectrum of behaviors (both sub-optimal and optimal). This is a positive outcome, not a failure. It proves that GAC has learned the complete data distribution without suffering from mode collapse, which is a prerequisite for effective posterior planning.
>
> - **Ablation experiments**
>
> Below we provide results on the Hopper task for the ablation. With other configurations unchanged, there is an trend that our architectural design and inference strategy lead to a lower training loss, and correspondingly, a better performance.
>
> | Task | Metric | GAC | No split $z$ | No $s_0$ | Langevin | LPT |
> | :--- | :--- | :--- | :--- | :--- | :--- | :--- |
> | **h-m** | Final Loss $\downarrow$ | **0.06482** | 0.07056 | 0.07246 | 0.13543 | 0.15638 |
> | | GAC-E[y] $\uparrow$ | **67.6 $\pm$ 5.2** | 53.8 $\pm$ 4.9 | 54.7 $\pm$ 7.1 | 52.7 $\pm$ 6.1 | 49.1 $\pm$ 9.1 |
> | | GAC-p(y+) $\uparrow$ | 63.1 $\pm$ 8.3 | 52.53 $\pm$ 6.5 | 44.4 $\pm$ 7.1 | 51.7 $\pm$ 8.4 | 50.2 $\pm$ 10.7 |
> | | | | | | | |
> | **h-m-r** | Final Loss $\downarrow$ | **0.11909** | 0.12073 | 0.12462 | 0.20486 | 0.21667 |
> | | GAC-E[y] $\uparrow$ | **83.4 $\pm$ 6.6** | 71.6 $\pm$ 16.6 | 58.4 $\pm$ 22.5 | 45.6 $\pm$ 12.9 | 63.3 $\pm$ 13.5 |
> | | GAC-p(y+) $\uparrow$ | 62.0 $\pm$ 22.3 | 63.7 $\pm$ 20.8 | 33.1 $\pm$ 28.6 | 45.2 $\pm$ 24.0 | 50.8 $\pm$ 17.8 |
>
> While the ablation does echo the reviewer's postulation that the architectural modifications would downgrade LPT model's performance, this comparision also shows that the new architecture is more compatible with GAC's VB training.
>
> - Q1. Line 189 means $p_\alpha(z)$. We've corrected it.
> - Q2. We've added a table summarizing differences between LPT and GAC in appendix.
> - Q3. How fast is this approach?
>
> Inference efficiency: While GAC incurs an initial planning overhead compared to model-free baselines, policy execution speed remains identical. This is similar to the CoT thinking in LLMs these days. Training efficency: As shown below, training completes in under 6 hours even with a single global update per inferred $z$, with further efficiency gains achievable via more frequent outer-loop optimizations. Generally speaking, [1] has shown signs of better training effiency of such VI training than standard LLMs.
>
> | datasets | Total time(s) | Each Inference(s) | Each Outer-loop(s) | Final loss |
> | :--- | :--- | :--- | :--- | :--- |
> | h-m outer-1 | 19548.1 | 31.0 | 0.1 | 0.06393 |
> | h-m outer-5 | 3807.4 | 29.8 | 1.0 | 0.06499 |
> | h-m outer-10 | 2076.7 | 31.9 | 2.1 | 0.06482 |
> | h-m outer-20 | 1038.1 | 31.0 | 4.3 | 0.06631 |
>
> - Q4. We've summarized the different algorithms in appendix.
> - Q5. We've added it.
> - Q6. We've added it in caption and standardize the $\delta$ in table.
> - Q7. We've corrected the typo.
> - Q8. Yes. In Figure 5, re-planning is triggered at waypoint B. We use the executed partial trajectory $\tau_{A \to B}$ as a condition to re-run the inference optimization (Eq. 3). This solves for a new latent plan $z'$ that maximizes the expected return $\mathbb{E}[y|z']$ while adhering to the history constraint via the posterior $p(z'|\tau_{A \to B})$.
>
> [1] Kong et al., Latent Thought Models with Variational Bayes Inference-Time Computation, ICML, 2025

---

> > ### Comment · Reviewer_UgEN · 2025-11-25
> >
> > I thank the authors for their thorough response and for addressing my questions. I maintain my positive assessment of the work and increase my confidence.

---

> > > ### Author Response · Authors · 2025-11-26
> > > **Response to Reviewer UgEN**
> > >
> > > We sincerely thank the reviewer for the continued support and for increasing the confidence. We are glad that our response addressed your questions.

---

### Official Review · Reviewer_s55t · 2025-10-31

**Soundness:** 3
**Presentation:** 4
**Contribution:** 2
**Rating:** 4
**Confidence:** 3

**Summary:**

This paper introduces Generative Actor-Critic (GAC), a new framework that formulates reinforcement learning as generative modeling and probabilistic inference over trajectory–return pairs. Instead of estimating expected returns, GAC learns a joint generative model $p(\tau,y)$ over trajectories $\tau$ and returns $y$ using a latent-variable model $p(z)p(\tau∣z)p(y∣z)$. Decision-making (both exploration and exploitation) is then performed via posterior inference on the latent plan variable $z$.

The approach allows effective offline-to-online RL adaptation, supports trajectory-level reasoning without step-wise rewards, and achieves state-of-the-art results on several continuous control and maze benchmarks.
- An LLM was used to improve writing.

**Strengths:**

1. Principled formulation — Viewing RL as generative inference offers a clear conceptual bridge between diffusion-style generative models and reinforcement learning.

2. Trajectory-level modeling — Operates on trajectory distributions rather than step-wise transitions, enabling sparse-reward settings.

3. Solid performance on standard benchmarks — Demonstrates strong offline-to-online adaptation even without step-level reward signals.

**Weaknesses:**

1. Outdated baselines

The experimental comparison primarily uses older O2O RL methods (e.g., ODT, Cal-QL, LPT). Missing comparisons with more recent approaches. This makes it difficult to judge whether improvements stem from the GAC formulation or simply from implementation differences.

2. Limited novelty / incremental over Decision Diffuser

Conceptually, GAC can be seen as a latent-variable extension of Decision Diffuser with a minor modification of the target formulation (the optimistic $y^+$). The “guided generation via optimistic target” resembles classifier-free guidance (CFG) and does not constitute a major conceptual departure. Lastly, the online fine-tuning stage essentially performs supervised fine-tuning (SFT) on new trajectories, rather than a fundamentally new RL adaptation mechanism.


3. Empirical scope

While GAC is evaluated on D4RL, evaluation is limited to low-dimensional control benchmarks, excluding kitchen dataset or adroit. Evaluation on mujoco is also excluding hopper, halfcheetah, and walker2d medium expert, which is one of the most common benchmark in d4rl mujoco.

**Questions:**

See weaknesses.

---

> ### Author Response · Authors · 2025-11-22
> **Response to Reviewer s55t (1/2)**
>
> We thank the reviewer for the detailed and critical feedback. We appreciate the opportunity to clarify the fundamental differences between GAC and prior works like Decision Diffuser. Your questions regarding the nature of our online adaptation and the baselines have helped us significantly sharpen the positioning of our contributions.
>
> **1. Outdated baseline?**
>
> While the field of (offline-to-online) RL is rapidly evolving, the underlying framework and principles are far more enduring. GAC is a new framework designed by first principles. This new framework started from a new angle, which is also a more challenging setting from the conventional viewpoint: utilizing only trajectory-level returns. Most recent SOTA methods fundamentally rely on dense step-wise rewards for Bellman updates, rendering them ineffective in our outcome-based focus. To illustrate this limitation, we evaluated the recent SOTA method SSAR [1] (specifically suggested by Reviewer SU1Q) under our return-only constraints ($R_{t<T}=0, R_T=y$). We use the SSAR public repo to evaluate MuJoCo datasets. As shown below, SSAR struggles significantly in this setting. This experiment serves as a crucial proof point: even state-of-the-art Bellman methods are incapable in sparse-return tasks.
>
> | Method | h-m | h-m-r | w-m | w-m-r | ha-m | ha-m-r |
> | :--- | :--- | :--- | :--- | :--- | :--- | :--- |
> | SSAR Cql backbone offline | 12.38 | 19.99 | 3.90 | 14.12 | 36.98 | -0.56 |
> | SSAR Td3+bc backbone offline | 7.35 | 20.27 | 0.27 | 25.48 | -1.64 | -1.28 |
> | GAC-E[y] offline | **67.6** | **83.4** | **80.2** | **78.9** | **43.6** | **39.8** |
> | SSAR Cql backbone online | 14.01 | 20.60 | 7.68 | 13.03 | 37.04 | 0.20 |
> | SSAR Td3+bc backbone online | 14.48 | 20.77 | 0.02 | 5.97 | -2.02 | 1.04 |
> | GAC-E[y] online | **94.9** | **96.8** | **85.1** | **85.4** | **44.3** | **40.9** |
>
> **2. Limited novelty? Incremental over Decision Diffuser?**
> The reviewer seems to have overlooked the core conceptual leap in this paper: While Decision Diffuser (DD) focuses on the conditional distribution $p(τ∣y)$, GAC models the complete joint distribution $p(τ,y)$. We hope our clarifications on their questions can help resolve this misunderstanding:
>
> - **GAC is an extension of Decision Diffuser?**
>
> GAC is a distinct framework with a fundamental shift in modeling paradigm. This distinction is crucial because it naturally entails modeling a critic $p(y∣τ)$, which enables us to move beyond passively **chasing** a fixed target return (as in DD) to performing active inference—specifically, **optimizing** the expected return or sampling from a dynamically adjusted posterior. In this context, the continuous latent plan $z$ is introduced not as a simple extension, but as a critical design choice to alleviate the computational cost of this inference. As demonstrated in Table 1, capability enabled by such inference (GAC−$E[y]$) consistently outperforms the fixed-target conditioning strategy (GAC−$y^*$, akin to DD).
>
> - **Guided generation via optimistic target resembles CFG?**
>
> While both GAC and DD-CFG employs the general idea of "guidance", the guidance in GAC is inherently aware of the prior distribution of trajectories and returns, while CFG is a heuristic sampling scheme that actually pushes the samples away from the prior. Empirically, while the fixed target $y^*$ is usually set to be the maximum value in the dataset, we kindly refer the reviewer to GAC's stability under OOD targets shown in the left 2 of Figure 4.
>
> - **GAC's online fine-tuning is SFT rather than RL adaptation?**
>
> Just because GAC can perform stably with OOD targets, our exploration strategy (GAC-$p(y^+)$) that dynamically sets the target can lead to samples better than what has been seen. As shown in left column of Fig 2, samples from GAC-$p(y^+)$ (the blue) always have better coverage at high-return regime (right end of x-axis) than fixed target CFG (GAC−$y^*$, the pink, akin to DD). This is the core promise of RL. In fact, this principled inference mechanism elegantly resolves the exploration-exploitation dilemma in classical RL, unveiling an unexplored power of "RL as generative inference". This adaptive exploration continuously and automatically enriches the training distribution with high-quality, diverse experiences, which in return improves the generative models, as shown in the middle column of Fig 2.
>
> [1] Luo et al., Learning to Trust Bellman Updates, arXiv preprint, 2025

---

> ### Author Response · Authors · 2025-11-22
> **Response to Reviewer s55t (2/2)**
>
> **3. Empirical scope**
>
> We thank the reviewer for this suggestion. We agree that GAC, as a general decision-making framework, is well-suited for future evaluation on more complex, high-dimensional tasks such as Adroit, Kitchen, or even vision-based tasks. This is an exciting next step for our work.
>
> But we also hope the reviewer would note that MuJoCo and Maze2D are two full-fledged tasks for evaluating models’ performance in high-dimensional control and long-range planning, respectively. This is recognized by reviewer R-UgEN, who praised our empirical study as “analyzing the proposed approach holistically on many aspects and does so very well with convincing evidence”. As for medium-expert dataset, we deliberately avoided it for a bigger headroom of models’ self-improvement, following Online Decision Transformer[2].
>
> [2] Zheng et al. Online Decision Transformer, ICML 2022.

---

> > ### Comment · Reviewer_s55t · 2025-11-27
> >
> > After reading the authors’ rebuttal, my understanding of the paper has improved, and their clarifications address most of my earlier concerns about novelty and the choice of baselines. While the empirical evaluation could be further strengthened by including more complex, high-dimensional benchmarks such as Adroit or Kitchen, the rebuttal convincingly resolves my main conceptual concerns, and I now have a more positive view of the work. I therefore increase my overall score to 6.

---

### Meta-Review · Area_Chair_c1WZ · 2026-01-07

**Summary:**

This paper introduces Generative Actor Critic (GAC), a framework that redefines Reinforcement Learning as a generative model of the joint trajectory-return distribution. While the approach is conceptually intriguing, there are concerns about its empirical rigor, baseline selection, and fairness of comparison. Despite the authors’ rebuttals, the paper fails to compare against contemporary state-of-the-art methods and employs an experimental setup that may inaccurately represent the performance of existing benchmarks.

**Reviewer Concerns:**

The paper omits comparisons with recent diffusion-based planners that are known to perform significantly better on benchmarks like Maze2D. For instance, the recent paper titled What Makes a Good Diffusion Planner for Decision Making? (ICLR 2025) included many powerful diffusion planner methods that share a similar modeling framework with the current paper. This omission is critical, as diffusion models are naturally suited for the trajectory-level reasoning and "stitching" that GAC claims as a primary strength.

The selection of Decision Transformer (DT) and IQL as primary baselines is outdated. There are more effective DT methods like Q-value Regularized Transformer for Offline Reinforcement Learning (ICML 2025) and other related works. Furthermore, the authors adapted these baselines by removing step-wise rewards, which is unfair as methods like IQL and CQL are explicitly designed to optimize via Bellman backups that require such signals. Effectively handicapping the baselines in this way makes the reported performance gains for GAC misleading. This further raises questions about whether the selection of baselines is proper.

Reviewers noted that GAC can be seen as an incremental extension of Decision Diffuser. While the authors argue that modeling the joint distribution enables "active inference", which partially addressed the reviewer’s question, the practical advantage over existing guided generation methods remains insufficiently demonstrated against strong, modern competitors due to the absence of such baseline methods.

**Reviewer Scores:**

Reviewer s55t: 4 (Raised to 6 in the comment section, though concerns about outdated baselines and limited empirical scope remain).

Reviewer UgEN: 8 (Maintained a positive assessment, focusing on the conceptual novelty of joint modeling).

Reviewer xAm5: 8 (Appreciated the latent-variable approach for exploration).

Reviewer SU1Q: 2 (Remained critical of the "unfair" baseline adaptation and the lack of quantitative evidence for latent space claims).

---

### Decision · Program_Chairs · 2026-01-26

Reject